# PROTOTYPES-INJECTED PROMPT FOR FEDERATED CLASS INCREMENTAL LEARNING

## ABSTRACT

Federated Class Incremental Learning (FCIL) is a new direction in continual learning (CL) for addressing catastrophic forgetting and non-IID data distribution simultaneously. Existing FCIL methods call for high communication costs and exemplars from previous classes along with performance issues. We propose a novel rehearsal-free method for FCIL named prototypes-injected prompt (PIP) that involves 3 main ideas: a) prototype injection on prompt learning, b) prototype augmentation, and c) weighted Gaussian aggregation on the server side. Our experiment result shows that the proposed method outperforms the current state of the arts (SOTAs) with a significant improvement $(14 - 33\%)$ in CIFAR100, MiniImageNet and TinyImageNet datasets. Our extensive analysis demonstrates the robustness of PIP in different task sizes, and the advantage of requiring smaller participating local clients, and smaller global rounds. For further study, source codes of PIP, baseline, and experimental logs are shared publicly in `https://anonymous.4open.science/r/an122pouyyt789/`.

## 1 INTRODUCTION

Federated learning (FL) is a machine learning approach that allows multiple local clients to learn a global model together while protecting the data privacy in each client McMahan et al. (2017)Karim-ireddy et al. (2020) Shoham et al. (2019) while protecting the data privacy in each client. FL has recently sparked a great deal of academic interest and achieved outstanding success in a number of application areas, including medical diagnosis Hwang et al. (2023), autonomous vehicle He et al. (2023), and wearable technology Baucas et al. (2023). The majority of FL methods are often designed for a static application scenario, assuming the data classes are fixed and known in advance. In real-world applications, however, the data are often dynamic, allowing local clients to access unseen target classes online.

Existing studies have the addressed dynamic data challenges in FL through Federated Class Incremental Learning (FCIL) where each local client gathers training data continually and according to their own preferences in the environment, while new clients with ad hoc, unforeseen classes are always welcome to join the FL training Dong et al. (2023) Dong et al. (2022) Yoon et al. (2021). The clients have to cooperatively train a global model to continually learn new classes while maintaining its capability to recognize the previous classes. In short, the existing FCIL methods tried to answer catastrophic forgetting and non-independently and identically distributed (non-i.i.d.) problems.

Current SOTAs in FCIL still have not achieved a high performance i.e. $leq$ 75% average accuracy in the 3 popular benchmark datasets Dong et al. (2023) Dong et al. (2022). the performance Besides the performance issue, current FCIL methods exchange the whole model between clients and the central server, resulting in high communication costs during FL training. Besides, some of the FCIL methods assume that the clients save exemplars (memory) from the previous classes for rehearsal Dong et al. (2023) Dong et al. (2022). In practice, this is not always feasible because the data may be confidential and only available for one-time access. Thus, we are motivated to develop a more efficient and rehearsal-free method for the FCIL problem. We propose a new approach for FCIL based on prompt learning inspired by prompt learning that performs effectively in stand-alone continual learning (CL) Wang et al. (2022c) Wang et al. (2022b). Along with the rehearsal-free principle, the prompt-based approach offers a more efficient communication where clients and the central server only need to exchange a small model called prompt and supporting parameters, e.g. head layer

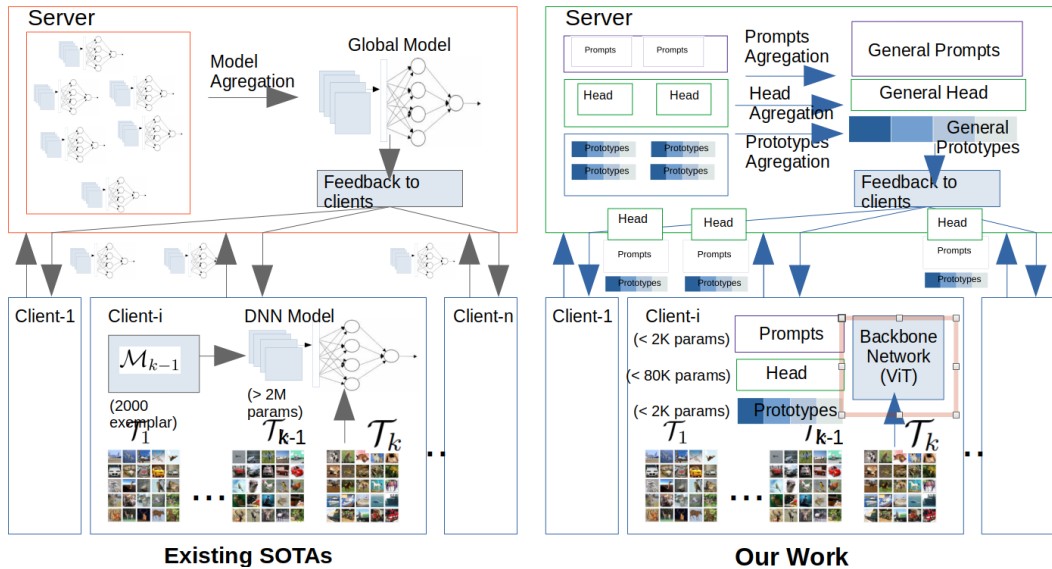

Figure 1: Visualization of current SOTAs in FCIL and our proposed method. It is seen that our approach imposes much lower communication costs than state-of-the-art methods.

instead of exchanging the whole model. The distinction between the PIP and current SOTAs is visualized in figure 1. **The contributions** of this paper are summarized as follows: (1) We propose a new approach, prompt-based federated continual learning to solve the FCIL problem, where clients and central servers exchange prompts and head layer units instead of a whole model. This approach significantly reduces the size of the exchanged parameters between clients and the central server in each round. (2) We propose a new baseline method for the FCIL problem named Federated-DualPrompt that already outperforms the current state-of-the-art (SOTA) methods in three benchmark datasets. (3) We propose a novel method for the FCIL problem named Prototype-Injected Prompt (PIP) that consists of three main ideas i.e. a) prototype injection on prompt learning, b) prototype augmentation, and c) weighted Gaussian aggregation on the server side. The proposed method outperforms the baseline and current SOTAs with a significant gap. (4) We provide a comprehensive analysis in three benchmark datasets as well as the robustness of the proposed method in different task sizes, smaller participating clients, and smaller rounds per task. (5) The source codes of PIP are shared in https://anonymous.4open.science/r/an122pouyyt789/ during the peer-review process and will be made public upon acceptance of our paper for further study and reproducibility.

## 2 RELATED WORK

This research relates to the recent study on **Class Incremental Learning (CIL)** which including: **a) Prompt-based approaches**, which trains a trainable set of parameters with a down-streaming approach for tasks sequence, instead of training the whole model e.g. L2P Wang et al. (2022c), DualPrompt Wang et al. (2022b), CODA-Prompt Smith et al. (2023), S-Prompt Wang et al. (2022a) **b) Regularization approaches**, which adaptively tune the base learner parameter to accommodate the previous task and current task e.g. EWC Kirkpatrick et al. (2017), SI Zenke et al. (2017), MAS Aljundi et al. (2018), LWF Li & Hoiem (2017), LWM Dhar et al. (2019). DMC Zhang et al. (2020). **c) Replay/Rehearsal approach**, which trains the exemplars (memory) from the previous task joined with current task data e.g. ICARL Rebuffi et al. (2017), EEIL, Castro et al. (2018), GD Prabhu et al. (2020), DER/DER++ Buzzega et al. (2020), XDER Boschini et al. (2022), **d) Bias correction approaches**, which creates task-wise bias later to help the base learner achieve plasticity on a new task while maintaining stability on old tasks e.g. BiC Wu et al. (2019), LUCIR Hou et al. (2019), iL2M Belouadah & Popescu (2019). A recent reserach on **Federated Learning (FL)** includes: FedAvg McMahan et al. (2017), Fedbn Li et al. (2021), FedMatchAvg Wang et al. (2020), Yang et al. (2021), Wang et al. (2021), Shoham et al. (2019), and a relatively new **Federated Class**

**Incremental Learning(FCIL)** problem includes: GLVC Dong et al. (2022), TARGET Zhang et al. (2023) and LGA Dong et al. (2023).

# 3 PRELIMINARIES

## 3.1 PROBLEM FORMULATION

A class incremental learning (CIL) problem is defined as a learning problem of a sequence of fully-supervised learning tasks $\mathcal{T}^1, \mathcal{T}^t, ..., \mathcal{T}^T$ where $t \in \{1, ..., T\}$ represents the number of tasks that is unknown. Each task carries $N_t$ pairs of training samples $\mathcal{T}^t = \{x_i^t, y_i^t\}_{i=1}^{N_t}$ where $x_i \in \mathcal{X}$ denotes input image and $y_i \in \mathcal{Y}$ denotes its corresponding class label. Each task carries the same image size but possesses disjoint target classes. Suppose that $Y^t$ labels a class set of a task $t$ and $Y^{t-1}$ denotes a class set of task $t - th$, $\forall t, t - 1, Y^t \cap Y^{t-1} = \emptyset$. Federated Class-Incremental Learning (FCIL) is the extension of conventional class-incremental learning in a federated way, where a set of clients collaborate with each other coordinated by a central server. Given local clients $\{S_l\}_{l=1}^L$ and a global central server $S_G$, for the $r - th$ global round ($r = 1, ..., R$), a set of local clients $S_l$ are randomly selected from all participating clients by the global server. The selected clients at the $r - th$ round simulate real-world conditions where only a small number of registered clients join in a round of federated learning. The clients may carry a different set of training samples $\mathcal{T}_1^{r,t} \neq \mathcal{T}_2^{r,t}... \neq \mathcal{T}_l^{r,t}$ where $l \in \{1, ..., L\}$, and $\mathcal{T}_l^{r,t} = \{x_{li}^{r,t}, y_{li}^{r,t}\}_{i=1}^{N_l^{r,t}}$ is not shareable between clients or between the client and central server. Technically, on each global round $r - th$ for the $t - th$ incremental task, a set of selected clients $\{S_l\}_{l=1}^L$ is selected, then each client $S_l$ conducts local CIL training using its training samples $\mathcal{T}_l^{r,t}$ to produce an optimal local model $\Theta_l^{r,t}$. Then the selected clients send their local models to global server $S_G$ to be aggregated, producing the latest global model parameter $\Theta_G^{r,t}$. Global server $S_G$ then distributes the global model to all clients for the next global round process.

## 3.2 FEDERATED CLASS INCREMENTAL LEARNING VIA PROMPT-BASED FEDERATED CONTINUAL LEARNING

Inspired by transfer learning in NLP, prompt-based learning was proposed in continual learning. In prompt-based continual learning, a client uses a frozen backbone model e.g. Vision Transformer (ViT) as a feature extractor. The ViT embedding layer transforms an input image $x \in R^{w \times h \times c}$, where $w, h, c$ are the image's width, height, and channel respectively, into sequence-like output feature $h \in R^{L \times D}$, where $L$ is the sequence length and $D$ is the embedding dimension. We extend the prompt-based continual learning into prompt-based federated continual learning for the FCIL problem. In this approach, the client only updates the learnable parameters called prompt $p \in R^{L_p \times D}$, and its classification (head) layer $f_\phi$, where $L_p$ is the sequence length and $D$ is the embedding dimension and $\phi$ is head layer parameter. In FCIL setting, on each global round $r - th$ for the $t - th$ task, the selected clients $\{S_l\}_{l=1}^L$ updates $p_l$ and $\phi_l$ parameters which are aggregated by global server $S_G$ producing global model parameters $p_G$ and $\phi_G$. In this approach, rather than updating the whole model, i.e. backbone and head layers, clients only update a far smaller amount of parameters, i.e. prompt and head layer only. In addition, clients and the central server require smaller communication costs as the clients and server exchange a smaller size of data on each round. The distinction between this proposed approach and the existing state-of-the-art (SOTA) methods is that the existing SOTAs update and aggregate the whole model i.e. $\Theta = (\theta, \phi)$ where a model $F_\Theta(x) = f_\phi(g_\theta(x))$, $g_\theta(.)$ is feature extractor and $f_\phi(.)$ is classifier.

## 3.3 BASELINE METHOD: FEDERATED DUALPROMPT

In this study, we propose a baseline method named Federated DualPrompt (Fed-DualPrompt) by customizing DualPrompt Wang et al. (2022b) for federated class incremental learning problems. **The role of the baseline** is to demonstrate our proposed new approach i.e. FCIL via prompt-based federated continual learning that is more effective and efficient than current SOTAs.

**Prompt-Structure:** Following DualPrompt structure Fed-DualPrompt uses 2 prompt types, i.e. G-Prompt and E-Prompt. G-Prompt is a trainable shared parameter for all tasks defined as $g \in R^{L_g \times D}$ where $L_g$ is the sequence length and $D$ is the embedding dimension. E-Prompt is a set of task-

specific parameters defined as $E = \{e^t\}_{t=1}^T$ where $e^t \in R^{L_e \times D}$, $L_e$ is the sequence length and $D$ is the embedding dimension.

**Clients:** On a round $r - th$ of $t - th$ task, each client $S_l$ with its available training samples $\mathcal{T}_l^{r,t}$ conducts local training to optimize the $t-th$ task E-prompt $e_l^t$, G-prompt $g_l$ and head layer parameter $\phi_l$ following Wang et al. (2022b). Given a pre-trained ViT backbone $f$ with $N$ Multi head Self-Attention (MSA), $h^{(i)}, i = 1, 2..., N$ represents input for $i - th$ MSA layer, suppose that we want to attach G-prompt $g_l$ and E-prompt $e_l^t$ into the $i - th$ and $j - th$ MSA layer respectively, then G-prompt $g_l$ and E-prompt $e_l^t$ transform feature $h^{(i)}$ and $h^{(j)}$ via prompt function as defined in equation 1. Note that $h^{(i)}$ and $h^{(j)}$ are the extension of $h$, a sequence-like parameter produced by the ViT embedding layer for $i - th$ and $j - th$ MSA layer.

$$(h_{g_l}^{(i)}, h_{e_l}^{(j)}) = (f_{prompt}(g_l, h^{(i)}), f_{prompt}(e_l^t, h^{(j)})) \tag{1}$$

Following Wang et al. (2022b), DualPrompt accommodates two types of function $f_{prompt}$ i.e. prompt tuningLester et al. (2021) and prefix tuning Li & Liang (2021). Previous study Wang et al. (2022b) investigated that prefix tuning is more effective for CIL problems, thus we use prefix tuning for $f_{prompt}$ for the proposed baseline method as defined in equation 2, where $p \in R^{L_p \times D}$ could be G-prompt $g_l$ and E-prompt $e_l^t$ that is split into $p_K \in R^{L_p/2 \times D}$ and $p_V \in R^{L_p/2 \times D}$.

$$f_{prompt}(p, h) = MSA(h_Q, [p_K; h_K], [p_V; h_V]) \tag{2}$$

MSA function is defined as in equation 3 following Vaswani et al. (2017) where $h_Q$, $h_K$, and $h_V$ are input query, key, and value, $W_Q$, $W_K$, and $W_V$ are the projection matrices, and $m$ is number of head, and in DualPrompt $h_Q = h_K = h_V = h \in R^{L \times D}$.

$$MSA(h_Q, h_K, h_V) = Concat(h_1, ..., h_m)W^O$$
$$where \; h_i = Attention(h_Q W_i^Q, h_K W_i^K, h_V W_i^V) \tag{3}$$

**Objective:** Given a pre-trained ViT backbone with attached G-prompt $g_l$ and E-prompt $e_l^t$, head layer $f_{\phi_l}$, task-wise key $k_l^t$ associated with $e_l^t$, the learning objective of each client $S_l$ is defined as in equation 4, where $\mathcal{L}$ represents cross-entropy loss, $\mathcal{L}_{match}$ represents matching loss, and $\lambda$ represents a constant factor.

$$\min_{g_l, e_l^t, k_l^t, \phi_l} \mathcal{L}(f_{\phi_l}(f_{g_l, e_l^t}(x), y)) + \lambda \mathcal{L}_{match}(x, k_l^t), x \in \mathcal{T}_l^{r,t} \tag{4}$$

**Server:** Central server $S_G$ coordinates local clients training and aggregates locally optimal parameters from selected local clients into a global model. The parameter aggregation for the baseline method is defined in the equation 5 where $g_l, e_l^t$ and $\phi_l$ are the G-prompt, E-prompt, and head layer parameter of client-$l$ respectively and $g_G, e_G^t$ and $\phi_G$ are the G-prompt, E-prompt and head layer parameter of global model respectively, and $L$ is the number of selected local clients. The pseudo-code of the baseline is presented in Algorithm 1 in Appendix A

$$(g_G, e_G^t, \phi_G) = \frac{1}{L} \sum_{l=1}^L (g_l, e_l^t, \phi_l) \tag{5}$$

## 4 PROPOSED METHOD: PROTOTYPE-INJECTED PROMPT (PIP)

**Prompt Structure and Function:** We adopt the prompt structure from the baseline method that uses G-prompt and E-prompt as defined in the previous subsection as well as the prompt function as defined in equation 1, 2 and 3. Our approach also incorporates other prompt functions i.e. prompt tuning Lester et al. (2021) and other prompt structures i.e. L2P Wang et al. (2022c) and CODA-PromptSmith et al. (2023).

**PIP ideas:** Given a baseline framework where on $r-th$ round of $t-th$ task, clients $\{S_l\}_{l=1}^L$ conducts continual learning with $\hat{\mathcal{T}}_l^{r,t} \subset \mathcal{T}_l^{r,t}$ where $|\hat{\mathcal{T}}_l^{r,t}| = \eta|\mathcal{T}_l^{r,t}|, \eta \in (0, 1)$ is the percentage of available classes in each client, and $|.|$ represents the number of elements, then the optimized parameters will be: $\hat{\phi}_l, \hat{g}_l, \hat{e_l^t} = \arg\min_{\hat{g}_l, \hat{e_l^t}, \hat{k_l^t}, \hat{\phi}} \mathbb{E}_{(\hat{x}, \hat{y}) \sim \hat{\mathcal{T}}_l^{r,t}} [\mathcal{L}(f_{\hat{\phi}_l}(f_{\hat{g}_l, \hat{e_l^t}}(\hat{x}), \hat{y})) + \lambda \mathcal{L}_{match}(\hat{x}, \hat{k_l^t})]$ only optimal for $\hat{\mathcal{T}}_l^{r,t}$ but not $\mathcal{T}_l^{r,t}$. Aggregating the prompts and head doesn't guarantee the models optimal for

$\mathcal{T}_l^{r,t}$ since in the next round the participating clients and their available data may be different from the current rounds' $\{S_l^{r,t}\}_{l=1}^L \neq \{S_l^{r+1,t}\}_{l=1}^L$, $\hat{\mathcal{T}}_l^{r+1,t} \neq \hat{\mathcal{T}}_l^{r,t}$. Furthermore, learning a label space into multiple steps e.g. rounds leads to catastrophic forgetting. Our first idea is to **a shareable prototype** sets $(W^t, Y_{W^t})$ and **injects** into prompt learning process that satisfies $\hat{\mathcal{T}}_l^{r,t} \cup (W^t, Y_{W^t}) \approx \mathcal{T}_l^{r,t}$, where $W^t$ is the prototype feature set, and $Y_{W^t}$ is the prototype label set. Therefore, the learning objective of each client satisfies $arg \min_{\hat{g}_l, \hat{e}_l^t, \hat{k}_l^t, \hat{\phi}} \mathbb{E}_{(\hat{x}, \hat{y}) \sim \hat{\mathcal{T}}_l^{r,t}} [\mathcal{L}(f_{\hat{\phi}_l}(f_{\hat{g}_l, \hat{e}_l^t}(\hat{x}) \cup W^t, \hat{y} \cup Y_{W^t})) +$
$\lambda \mathcal{L}_{match}(\hat{x}, \hat{k}_l^t)] \approx arg \min_{\hat{g}_l, \hat{e}_l^t, \hat{k}_l^t, \hat{\phi}} \mathbb{E}_{(x,y) \sim \hat{\mathcal{T}}_l^{r,t}} [\mathcal{L}(f_{\hat{\phi}_l}(f_{\hat{g}_l, \hat{e}_l^t}(x), y)) + \lambda \mathcal{L}_{match}(x, \hat{k}_l^t)]$. The prototype sharing ensures each client receives globally available labels gathered from all participating clients to learn, rather than its locally available labels as $W^t = W_1^t \cup W_2^t \cup ... \cup W_l^t \cup ... W_L^t \supseteq W_l^t$ for $l \in \{1...L\}$. Our second idea i.e. **prototype augmentation** handles imbalance class between locally available classes and unavailable classes that are represented by the shared prototypes. Suppose that $x_{c1}$ is the sample for class $c1 \in C1$, $w_{c2}^t \in W^t$ is the prototype of class $c2 \in C2$, where $C1$ and $C2$ are the available and unavailable classes in client-$l$, we have $|x_{c1}| = N_{c1} \gg 1$, while $|w_{c2}^t| = 1$. The prototype augmentations create artificial prototypes that satisfy $|w_{c2}^t| \approx |x_{c1}|$. Our third idea i.e. **server weighted Gaussian aggregation** treats clients' contribution in global aggregation proportionally based on their participation and the number of training samples. It is based on the best practice that the model that learns more, has weight more convergence rather than a newly participating model. We view clients as observers of their local data and the parameters they produce as Gaussian distributions.

**Clients:** Similar to the baseline method, given a pre-trained ViT model $f$, with $N$ MSA layers, G-prompt $g_l$ and E-prompt $E_l = \{e_l^t\}_{t=1}^T$, and head layer $f_\phi$, each client $S_l$ conduct a local CIL training on $r-th$ round of $t-th$ task with its training samples $\mathcal{T}_l^{r,t}$. The distinction between the proposed method and the baseline is that the proposed method generates a set of prototypes $W^t$ and utilizes it to optimize $g_l$, $e_l^t$, and $\phi_l$. Later on, the prototype set is aggregated by the central server and returned to clients so that the clients receive a more completed and generalized prototype set.

**Prototype:** We define a set of prototypes on the $t-th$ task on a client $S_l$ as $W_l^t = \{w_{lc}^t\}_{c=a}^{c=b}$, $w_{lc}^t \in R^{1 \times D}$, and their labels as $Y_{W_l^t} = \{c\}_{c=a}^{c=b}$ where $[a,b]$ is the available classes in $\mathcal{T}_l^{r,t}$, and $D$ is the embedding dimension. Assuming that the prototype follows a Gaussian distribution $w_{lc}^t \sim \mathcal{N}(\mu_{lc}^t, \Sigma_{lc}^t)$ and the prototype is considered as $D$ disjoint uni-variate distribution, then we have $w_{lc}^t \sim \mathcal{N}(\mu_{lc}^t, \sigma_{lc}^t{}^2)$ where $\sigma_{lc}^t{}^2 = I_D.\sigma_{lc,i}^t{}^2, i \in \{1, 2..D\}$. We compute the prototype $w_{lc}^t$ properties by equations 6 and 7.

$$\mu_{lc}^t = \frac{1}{\sum_{li=1}^{N_l} 1 \, if(y_{li} = c)} \sum_{li=1}^{N_l} f_{g_l, e_l^t}(x_{li}) \, if(y_{li} = c) \tag{6}$$

$$\sigma_{lc}^t{}^2 = \frac{1}{\sum_{li=1}^{N_l} 1 \, if(y_{li} = c)} \sum_{li=1}^{N_l} (\mu_c^t - f_{g_l, e_l^t}(x_{li}))^2 \, if(y_{li} = c) \tag{7}$$

**Prototype augmentation:** We can simply assign $w_{lc}^t$ by $\mu_{lc}^t$ to generate a single prototype for class-c. However, we enrich the prototypes by using an augmentation as defined in equation 8 to generate $m$ augmented prototypes for class-c based on Gaussian distribution $w_{lc}^t \sim \mathcal{N}(\mu_{lc}^t, \sigma_{lc}^t{}^2)$, and $\beta$ is a random value in $(0,1)$ range.

$$w_{lc}^t = \{\mu_{lc}^t\} \cup \{u_{lci}^t\}_{i=1}^{i=m}, \, where \, u_{lci}^t = \mu_{lc}^t + \beta \sigma_{lc}^t, m = 1..5 \tag{8}$$

**Objective:** We extend the baseline learning mechanism by injecting the set of prototypes $W^t$ and its label set $Y_{W^t}$ into prompt-generated features $f_{g_l, e_l^t}(x)$ before patching into the head layer $f_\phi$. Therefore, the learning objective of each client is defined as in equation 9. In the first round of federated learning, each client has a prototype set from its available classes $W^t = W_l^t$, but after the aggregation process, it utilizes the aggregated prototype set $W^t = W_G^t \supseteq W_l^t$,

$$\min_{g_l, e_l^t, k_l^t, \phi_l} \mathcal{L}(f_{\phi_l}(f_{g_l, e_l^t}(x) \cup W^t, y \cup Y_{W^t})) + \lambda \mathcal{L}_{match}(x, k_l^t), x \in \mathcal{D}_t \tag{9}$$

**Server weighted aggregation:** We define the weight of a client-l on the t-th task as $\omega_l = \rho_{t,l} N$, where $\rho_t$ is the total of client-l participation until t-th task and $N = |\mathcal{T}_l^{r,t}|$ is the number of samples

that available in the client-l at that time. We design a Gaussian-based weighted aggregation as defined in equations 10 and 11. The derivation of the proposed weighted aggregation in presented in Appendix B. The pseudo-code of PIP is presented in Algorithm 2 in Appendix C

$$(g_G, e_G^t, \phi_G) = \frac{1}{\sum_{l=1}^{L} \omega_l} \sum_{l=1}^{L} (g_l \omega_l, e_l^t \omega_l, \phi_l \omega_l) \tag{10}$$

$$w_{Gc}^t = \mathcal{N}(\mu_{Gc}^t, \sigma_{Gc}^t{}^2), \ where$$

$$\mu_{Gc}^t = \frac{1}{\sum_{l=1}^{L} a_{lc}^t \omega_l} \sum_{l=1}^{L} a_{lc}^t \mu_{lc}^t \omega_l$$

$$\sigma_{Gc}^t{}^2 = \frac{1}{\sum_{l=1}^{L} a_{lc}^t \omega_l} \sum_{l=1}^{L} a_{lc}^t (\mu_{lc}^t{}^2 + \sigma_{lc}^t{}^2) \omega_l - \mu_{Gc}^t{}^2 \tag{11}$$

$$a_{lc}^t = 1, \ if(\mu_{lc}^t \ exists) \ else \ = 0$$

## 5 EXPERIMENT RESULTS AND ANALYSIS

### 5.1 EXPERIMENTAL SETTING

**Datasets:** Our experiment is conducted using three main benchmarks in FCIL i.e. split CIFAR100, split MiniImageNet, and split TinyImageNet. The CIFAR100 and miniImageNet datasets each contain 100 classes while TinyImagenet is a dtaset with 200 classes. We follow settings from Dong et al. (2022) and Dong et al. (2023) where the dataset is split equally into all tasks. In our main numerical result, The dataset is split into 10 tasks i.e. 10 classes per task for CIFAR100 and MiniImageNet, and 20 classes per task for the TinyImageNet dataset. In our further analysis, we investigate the performance of the proposed methods in different task sizes e.g. T=5 and T=20.

**Benchmark Algorithms:** PIP is compared with 10 state-of-the-art algorithms: LGA Dong et al. (2023), GLFCDong et al. (2022), AFCKang et al. (2022)+FL, DyToxDouillard et al. (2022)+FL, SS-ILAhn et al. (2021)+FL, GeoDLSimon et al. (2021)+iCaRLRebuffi et al. (2017)+FL, DDEHu et al. (2021)+iCaRLRebuffi et al. (2017)+FL, PODNetDouillard et al. (2020)+FL, BiCWu et al. (2019)+FL, iCaRLRebuffi et al. (2017)+FL, and the proposed baseline model (Fed-DualPrompt)

**Experimental Details:** Our numerical study is executed under a single NVIDIA A100 GPU with 40 GB memory across 3 runs with different random seeds {2021,2022,2023}. PIP and Fed-DualPrompt use pre-trained ViT as the backbone network, while the competitors use LeNet LeCun et al. (2015) as the feature extractor. Following Dong et al. (2023) and Dong et al. (2022), each experiment is simulated by 30 total clients and 1 global server, where in each round, 10 (33.33%) local clients are selected randomly. Each client randomly receives 60% ($\eta = 0.6$) class label space. The total global round is set to 100, the local clients' epoch is set to 2, and the learning rate is set by choosing the best value from {0.02,0.002}.

### 5.2 NUMERICAL RESULTS

The numerical result of the consolidated algorithms is summarized in table 1. The complete numerical result is shown in tables 4, 5, and 6 in Appendix D. The baseline method already achieves higher average accuracy (Avg) than the SOTA methods with $1-13\%$ improvement in accuracy. The baseline also experiences a lower performance drop (PD) $(19-27\%)$ compared to the SOTA methods $(\geq 26\%)$ except in the CIFAR100 dataset vs. LGA. The proposed method (PIP) achieves the highest accuracy with $\geq 10\%$ gap compared to the baseline method, and $\geq 14\%$ gap compared to the competitor methods. The proposed method also achieves the lowest performance drop with $(10-13\%)$ gap compared to the baseline and $\geq 12\%$ gap compared to the competitor methods. The table shows that PIP achieves a higher gap in TinyImageNet which is a relatively more complex problem than in MiniImageNet and CIFAR100.

Looking at per-task performance, tables 4, 5, and 6 show that PIP achieves the highest accuracy in all tasks in those three datasets. In the first task, the proposed method achieves higher performance with a small gap $1-4\%$ gap compared to the baseline method. However, with the increasing number

of tasks, the gap gets higher e.g. $5 - 12\%$ in task-2, $5 - 20\%$ in task-3, and $6 - 20\%$ in task-10 (last task). It shows that the proposed method handles catastrophic forgetting better than the baseline method.

| Method | CIFAR100 | | | MiniImageNet | | | TinyImageNet | | |
|---|---|---|---|---|---|---|---|---|---|
| | Avg | PD | Imp | Avg | PD | Imp | Avg | PD | Imp |
| iCaRL+FL | 51.79 | 52.30 | 36.21 | 46.46 | 41.30 | 44.56 | 40.14 | 35.00 | 46.12 |
| BiC+FL | 55.01 | 48.00 | 32.99 | 49.03 | 40.30 | 41.99 | 42.13 | 36.30 | 44.13 |
| PODNet+FL | 59.66 | 44.00 | 28.34 | 52.56 | 36.00 | 38.46 | 44.60 | 35.40 | 41.66 |
| DDE+iCaRL+FL | 58.79 | 43.70 | 29.21 | 52.30 | 37.00 | 38.72 | 44.46 | 37.70 | 41.80 |
| GeoDL+iCaRL+FL | 61.52 | 40.70 | 26.48 | 49.90 | 37.00 | 41.12 | 44.63 | 36.30 | 41.63 |
| SS-IL+FL | 54.90 | 44.30 | 33.10 | 46.43 | 32.40 | 44.59 | 37.90 | 35.00 | 48.36 |
| DyTox+FL | 65.35 | 34.10 | 22.65 | 50.30 | 41.00 | 40.72 | 45.43 | 42.60 | 40.83 |
| AFC+FL | 58.18 | 42.00 | 29.82 | 53.42 | 46.40 | 37.60 | 41.95 | 46.60 | 44.31 |
| GLFC | 66.92 | 40.00 | 21.08 | 56.99 | 30.30 | 34.03 | 47.89 | 31.00 | 38.37 |
| LGA | 73.45 | 26.70 | 14.55 | 67.51 | 25.50 | 23.51 | 53.18 | 33.00 | 33.08 |
| Fed-DualPrompt | 74.23 | 27.06 | 13.77 | 80.14 | 19.07 | 10.88 | 66.39 | 24.98 | 19.87 |
| **PIP(Ours)** | **88.00** | **14.39** | **-** | **91.02** | **9.23** | **-** | **86.26** | **11.36** | **-** |

Table 1: Summarized numerical results on CIFAR100, MniImageNet, and TinyImageNet (T=10), "Avg" denotes the average accuracy of all tasks, "PD" denotes performance drop, and "Imp" denotes improvement/gap of PIP compared to the respective method

## 5.3 ABLATION STUDY

We conducted an ablation study to investigate the contribution of each component of the proposed method. The result is summarized in Table 2, while the detailed result is presented in Appendix F. The result shows that the prototype and augmentation contribute the most to the improvement of the performance as shown by the performance difference of configurations E vs. F (12%), and G vs. H (16%). The weighted aggregation improves performance up to 1% as shown by the performance difference of PIP and configuration F. The head layer aggregation also plays an important role in the proposed method as shown that the absence of this component decreases performance with $\geq 6\%$ margin. The absence of two components e.g. prototype and head aggregation (configuration C) or prototype and weighted aggregation (configuration E) impact significantly the performance with 22% and 13% accuracy drop respectively. The absence of three components i.e. prototype, weighted aggregation, and head aggregation (configuration A) cost a 23% performance drop in the method.

| Config. | Prompt | Proto+Aug | W.Agg | Head | T1 (C=10) | T10 (C=100) | Avg | PD | Imp |
|---|---|---|---|---|---|---|---|---|---|
| A | ✓ | | | | 82.15 | 60.66 | 65.69 | 21.49 | 22.59 |
| B | ✓ | ✓ | | | 90.99 | 77.56 | 81.59 | **13.43** | 6.69 |
| C | ✓ | | ✓ | | 82.13 | 61.35 | 66.10 | 20.78 | 22.18 |
| D | ✓ | ✓ | ✓ | | 91.37 | 78.47 | 82.41 | **12.90** | 5.87 |
| E | ✓ | | | ✓ | 96.60 | 70.68 | 75.38 | 25.92 | 12.90 |
| F | ✓ | ✓ | | ✓ | **98.90** | 84.17 | 87.82 | 14.73 | 0.46 |
| G | ✓ | | ✓ | ✓ | 98.60 | 69.94 | 72.57 | 28.66 | 15.71 |
| **PIP** | ✓ | ✓ | ✓ | ✓ | **98.60** | **84.60** | **88.28** | **14.00** | **-** |

Table 2: Ablation study on CIFAR100 dataset in one-seeded run i.e. 2021. "Avg" denotes the average accuracy of all tasks, "PD" denotes performance drop, and "Imp" denotes improvement/gap of PIP compared to the respective configuration

## 5.4 FURTHER ANALAYIS

**a) Different task size:** We evaluate the performance of the proposed method compared to the competitor methods in different task size i.e. T=5 and T=20 to further investigate the robustness of the proposed method. Figure 2 summarizes the performance of the consolidated methods in CIFAR100, MiniImageNet, and TinyImageNet with T=5 (upper figures) and T=20 (bottom figures). Both in T=5 and T=20 settings, PIP achieves the highest performance in every task. Besides, all figures show that the proposed method has gentle slopes compared to the baseline and competitor methods.

It shows that the proposed method experiences the lowest performance drop from $t_{th}$ to $t + 1_{th}$ task. It confirmed the robustness of the proposed method in different task-size settings. The baseline (Fed-DualPrompt) method achieves better accuracy than the competitor methods in all 6 settings, except in CIFAR100 with the T=5 setting. In CIFAR100 with the T=5 setting, the baseline method achieves higher performance in task-1 and task-5, but lower performance in task-3, and comparable performance in task-2 and task-4. It shows the promising idea of the federated prompt-based approach for the FCIL problem.

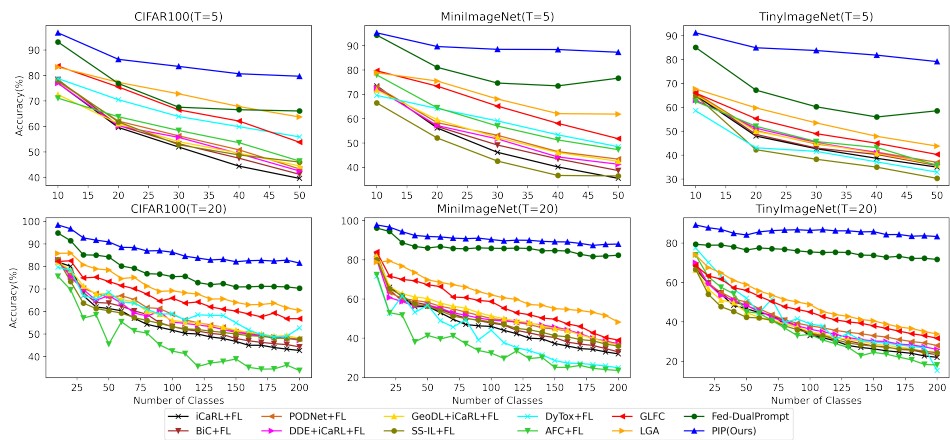

Figure 2: Performance of the consolidated methods in CIFAR100, MiniImageNet and TinyImageNet with T=5 and T=20

**b) Small local clients:** We evaluate the performance of the proposed method in smaller selected local clients to advance our investigation of the robustness of the proposed method on smaller selected local clients. This simulates cross-device federated learning where in a round only a small portion of registered clients participate in the federated learning process. In our experiment with 30 total clients, we investigate the performance of the consolidated methods with 3 (10%) and 2 (6.67%) selected clients compared with the default setting with 10 clients (33.33%). Figure 3 shows the performance of the consolidated methods in smaller local clients scenarios, while the complete result is presented in Appendix H. The figure shows that the proposed method experiences the lowest performance degradation in a smaller number of local clients. The figures show that both in CIFAR100 and TinyImageNet datasets, the proposed method still achieves higher accuracy than the baseline and the competitor methods with 10 participating local clients.

**c) Small global rounds:** We continue the previous experiment in smaller selected local clients into smaller local clients with smaller global rounds to study the robustness of the proposed method on a more extreme condition. This scenario simulates a condition where the model is urgently needed by the clients. In our experiment, we use 30 total clients and 3 (10%) local clients. Figure 4 summarizes our investigation on the performance of the proposed methods with 20 to 100 (default) global rounds in CIFAR100 and TinyImageNet datasets, while the complete result is presented in Appendix I. The figure shows a common trend that the performance of the method is decreasing with the decreasing of global rounds. However, our proposed method (PIP) experiences a relatively small amount of performance drop with the decreasing global rounds. Meanwhile, the baseline method experiences a significant drop in the CIFAR100 dataset when the number of global rounds is reduced to 20. Besides, It may experience over-fitting when the number of global rounds is too high as shown by the result on the TinyImageNet dataset. Compared to the current SOTA (GLFC and LGA) methods with the default number of global round (100), our proposed method (PIP) still achieve higher performance, even though the number of global rounds is reduced to 20 (20% default setting).

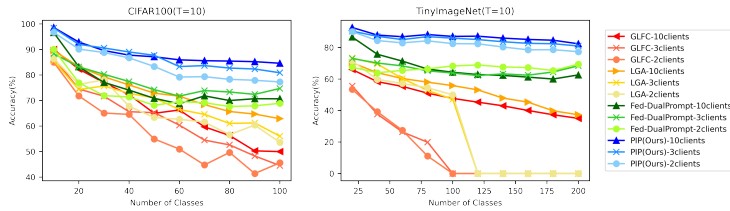

Figure 3: Performance of the consolidated methods in CIFAR100 and TinyImageNet with smaller local clients

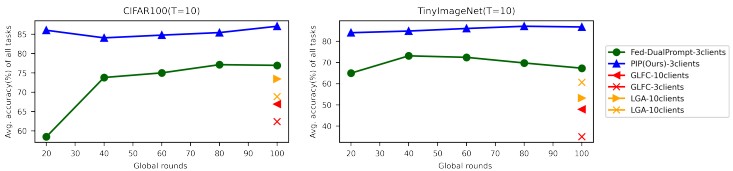

Figure 4: Performance of the consolidated methods in CIFAR100 and TinyImageNet with smaller local clients and smaller rounds

## 5.5 COMPLEXITY AND RUNNING TIME ANALYSIS

We evaluate the complexity of the proposed method as well as the complexity of the baseline method. Our complexity analysis shows that both the baseline method and our proposed method has the same complexity i.e. $O(R.L.N)$ where $R$ is the number of global round, $L$ is the number of participating local clients in each round, and $N$ is the size of the training data in each client. The detailed complexity analysis is provided in Appendix E. Table 5.5 summarizes the training time of the consolidated method in three datasets with T=10. The table shows that the proposed method requires lower training time than the current SOTA methods in all datasets. The training time is higher than the baseline training time because in our proposed method there are additional processes that do not exist in the baseline i.e. prototype generation, augmentation, injection (concatenation with prompt features), aggregation, and feedback.

| Method | CIFAR100 | MiniImageNet | TinyImagenet |
|---|---|---|---|
| GLFC | 24.3h | 36.6h | 46.4h |
| LGA | 23.2h | 35.7h | 45.9h |
| Fed-DualPrompt | 11.3h | 11.1h | 19.5h |
| PIP(Ours) | 13.0h | 14.5h | 25.5h |

Table 3: Training time of the consolidated algorithm in CIFARR100, MiniImagenet and TinyImageNet

## 6 CONCLUDING REMARKS

In this paper, we propose a new approach named prompt-based federated learning, a new baseline named Fed-DualPrompt, and a novel method named prototype-injected prompt (PIP) for the FCIL problem. PIP consists of three main ideas: a) prototype injection on prompt, b) prototype augmentation, and c) weighted Gaussian aggregation on the server side. Our experimental result shows that the proposed method outperforms the current SOTAs with a significant gap $(14-33\%)$ in CIFAR100, MiniImageNet, and TinyImageNet datasets. Our extensive analysis demonstrates the robustness of our proposed method in different task sizes, smaller participating local clients, and smaller global rounds. Our proposed method has the same complexity as the baseline method and experimentally requires shorter training time than the current SOTAs. In practice, our proposed method can be applied in both cross-silo and cross-domain federated class incremental learning.

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

## A  BASELINE ALGORITHM

Here we present the detailed algorithm of the proposed baseline in Algorithm 1.

---

**Algorithm 1** Fed-DualPrompt

---
1: **Input:** Number of clients $N$, number of selected local clients $L$, total number of rounds $R$, number of task $T$, local epochs $E$, batch size $B$.
2: Distribute frozen ViT backbone $f$ to all clients $\{S_l\}_{l=1}^{N}$ and central server $S_G$
3: Initiate G-prompt, E-prompt, and head layer for all clients and central server $g_G = g_l$, $E_G = \{e_G^t\}_{t=1}^{t=T} = E_l = \{e_l^t\}_{t=1}^{t=T}$, $\phi_G = \phi_l$, $l \in \{1..L\}$
4: $R_T \leftarrow R/T$, $R_T$ represents round per task
5: **for** $t = 1 : T$ **do**
6:     **for** $r = 1 : R_T$ **do**
7:         $S_l \leftarrow$ randomly select $L$ local clients from $N$ total clients
8:         **Clients execute**:
9:         Receive global parameters i.e. prompt, key, and head layer $g_G$, $e_G^t$, and $\phi_G$
10:         Assign local parameters $(g_l, e_l^t, \phi_l) \leftarrow (g_G, e_G^t, \phi_G)$
11:         $\mathcal{B} \leftarrow$ Split $\mathcal{T}_l^{r,t} = \mathcal{T}_l^t$ into $B$ sized batches
12:         **for** $e = 1 : E$ **do**
13:             **for** $b = 1 : \mathcal{B}$ **do**
14:                 Compute prompt-generated feature $f_{g_l, e_l^t}(x)$ as in Eq.1 to 3
15:                 Compute logits $f_{\phi_l}(f_{g_l, e_l^t}(x))$
16:                 Compute loss $\mathcal{L}_{total} = \mathcal{L} + \mathcal{L}_{match}$ as in Eq.4
17:                 Update local parameters $(g_l, e_l^t, \phi_l)$ based on $\mathcal{L}_{total}$
18:             **end for**
19:         **end for**
20:         Store local parameters $(g_l, e_l^t, \phi_l)$
21:         Send local parameters $(g_l, e_l^t, \phi_l)$ to server
22:         **Server executes**:
23:         Receives selected local clients $S_l$ parameters $(g_l, e_l^t, \phi_l)$
24:         Aggregates clients' parameters into global parameters $(g_G, e_G^t, \phi_G) \leftarrow (g_l, e_l^t, \phi_l)$ Eq5
25:         Send global parameters $(g_G, e_G^t, \phi_G)$ to clients for the next round
26:     **end for**
27: **end for**
28: **Output:** Global parameters $(g_G, e_G^t, \phi_G)$ and local parameters $(g_l, e_l^t, \phi_l), l \in 1..N$

---

## B  DERIVATION OF WEIGHTED AGGREGATION

Suppose that we have $n$ samples of an observation $x_i$ with the weight of $w_i$. The we mean and variance as:

$$\mu = \frac{1}{\sum_{i=1}^{n} w_i} \sum_{i=1}^{n} (x_i . w_i) \tag{12}$$

$$\sigma^2 = \frac{1}{\sum_{i=1}^{n} w_i} \sum_{i=1}^{n} w_i (x_i - \mu)^2 \approx \frac{1}{\sum_{i=1}^{n} w_i} \sum_{i=1}^{n} (w_i x_i^2 - w_i \mu^2) \tag{13}$$

Or we have

$$\sum_{i=1}^{n} w_i \sigma^2 = \sum_{i=1}^{n} (w_i x_i^2 - w_i \mu^2) = \sum_{i=1}^{n} w_i x_i^2 - \sum_{i=1}^{n} w_i \mu^2 \tag{14}$$

that equal

$$\sum_{i=1}^{n} w_i \sigma^2 + \sum_{i=1}^{n} w_i \mu^2 = \sum_{i=1}^{n} w_i x_i^2 \tag{15}$$

If we have another $m$ observation, then we have

$$\sum_{i=1}^{n+m} w_i \sigma^2 + \sum_{i=1}^{n+m} w_i \mu^2 = \sum_{i=1}^{n+m} w_i x_i^2 \tag{16}$$

$$\sum_{i=1}^{n+m} w_i \sigma^2 + \sum_{i=1}^{n+m} w_i \mu^2 = \sum_{i=1}^{n} w_i x_i{}^2 + \sum_{i=n+1}^{n+m} w_i x_i{}^2 \tag{17}$$

$$\sum_{i=1}^{n+m} w_i \sigma^2 + \sum_{i=1}^{n+m} w_i \mu^2 = (\sum_{i=1}^{n} w_i \sigma_{1:n}{}^2 + \sum_{i=1}^{n} w_i \mu_{1:n}{}^2) + (\sum_{i=n+1}^{n+m} w_i \sigma_{n+1:m}{}^2 + \sum_{i=n+1}^{n+m} w_i \mu_{n+1:m}{}^2) \tag{18}$$

$$\sum_{i=1}^{n+m} w_i \sigma^2 + \sum_{i=1}^{n+m} w_i \mu^2 = \sum_{i=1}^{n} w_i (\sigma_{1:n}^2 + \mu_{1:n}{}^2) + \sum_{i=n+1}^{n+m} w_i (\sigma_{n+1:m}^2 + \mu_{n+1:m}{}^2) \tag{19}$$

Therefore:

$$\sigma^2 = \frac{\sum_{i=1}^{n} w_i (\sigma_{1:n}^2 + \mu_{1:n}{}^2) + \sum_{i=n+1}^{n+m} w_i (\sigma_{n+1:m}^2 + \mu_{n+1:m}{}^2) - \sum_{i=1}^{n+m} w_i \mu^2}{\sum_{i=1}^{n+m} w_i} \tag{20}$$

$$\sigma^2 = \frac{\sum_{i=1}^{n} w_i (\sigma_{1:n}^2 + \mu_{1:n}{}^2) + \sum_{i=n+1}^{n+m} w_i (\sigma_{n+1:m}^2 + \mu_{n+1:m}{}^2)}{\sum_{i=1}^{n+m} w_i} - \mu^2 \tag{21}$$

The derivation above shows that if we have two weighted Gaussian distributions e.g. $X_1 \sim \mathcal{N}(\mu_1, \sigma_1{}^2)$ and $X_2 \sim \mathcal{N}(\mu_2, \sigma_2^2)$ with total weight $W_1$ and $W_2$ respectively, then the aggregated distribution will be:

$$\mu* = \frac{(\mu_1.W_1 + \mu_2.W_2)}{W_1 + W_2} \tag{22}$$

$$\sigma*^2 = \frac{((\mu_1{}^2 + \sigma_1{}^2).W_1 + (\mu_2{}^2 + \sigma_2{}^2).W_2)}{W_1 + W_2} - \mu*^2 \tag{23}$$

Generalizing equations above into $N$ observations i.e. $X_1 \sim \mathcal{N}(\mu_1, \sigma_1{}^2)$, $X_2 \sim \mathcal{N}(\mu_2, \sigma_2{}^2)$, ... $X_N \sim \mathcal{N}(\mu_N, \sigma_N{}^2)$ with total weight $W_1, W_2, ... W_N$ respectively. then the aggregated distribution will be: s

$$\mu* = \frac{\sum_{i=1}^{N} (\mu_i.W_i)}{\sum_{i=1}^{N} W_i} \tag{24}$$

$$\sigma*^2 = \frac{\sum_{i=1}^{N} (\mu_i{}^2 + \sigma_i{}^2).W_i}{\sum_{i=1}^{N} W_i} - \mu*^2 \tag{25}$$

## C  PIP ALGORITHM

Here we present the detailed algorithm of our proposed method in Algorithm 2.

## D  COMPLETE NUMERICAL RESULT ON 3 BENCHMARK DATASETS (T=10)

## E  COMPLEXITY ANALYSIS

In this section, we analyze the complexity of the baseline and the proposed method. Suppose that $N_l$ is the total samples of a dataset of a client across all tasks, $t \in \{1, ..., T\}$ is task index, $N_l^t = |\mathcal{T}_l^{r,t}| = |\mathcal{T}_l^t|$ is the number of samples on task $t$ in client-$l$ that satisfy $\sum_{t=1}^{T} N_l^t = N_l$, $R$ is the total rounds of federated learning, $E$ is the number of local epoch for each client training. $\beta$ is the number of batches on each task that satisfy $\sum_{b=1}^{\beta} N_{bl}^t = N_l^t$. We simplify the derivation by analyzing the complexity in a common case that the tasks are divided evenly, therefore we have $|\mathcal{T}_l^1| = |\mathcal{T}_l^2|... = |\mathcal{T}_l^t|... = |\mathcal{T}_l^T|$, that equal $N_l = T.N_l^t = T.|\mathcal{T}_l^t|$. Let $O(.)$ denote the complexity of a process.

**Baseline Complexity:** Following the pseudo-code in Algorithm 1 then we have

$$O(Baseline) = O(1) + T.R_T.(O(clients) + O(server)) \tag{26}$$

**Algorithm 2** Algorithm 2: PIP

---

1: **Input:** Number of clients $N$, number of selected local clients $L$, total number of rounds $R$, number of task $T$, local epochs $E$, batch size $B$.
2: Distribute frozen ViT backbone $f$ to all clients $\{S_l\}_{l=1}^N$ and central server $S_G$
3: Initiate G-prompt, E-prompt, key, and head layer for all clients and central server $g_G = g_l$, $E_G = \{e_G^t\}_{t=1}^{t=T} = E_l = \{e_l^t\}_{t=1}^{t=T}$, $\phi_G = \phi_l$, $l \in \{1..L\}$
4: $R_T \leftarrow R/T$, $R_T$ represents round per task
5: **for** $t = 1 : T$ **do**
6:     Init global and local prototypes $W_G^t = W_l^t, = W^t = \emptyset$
7:     **for** $r = 1 : R_T$ **do**
8:         $S_l \leftarrow$ randomly select $L$ local clients from $N$ total clients
9:         **Clients execute**:
10:         Receive global parameters i.e. prompt, prototypes, and head layer $g_G$, $e_G^t$, $W_G^t$, and $\phi_G$
11:         Assign local parameters $(g_l, e_l^t, \phi_l, W_l^t) \leftarrow (g_G, e_G^t, \phi_G, W_G^t)$
12:         $\mathcal{B} \leftarrow$ Split $\mathcal{T}_l^{r,t} = \mathcal{T}_l^t$ into $B$ sized batches
13:         **for** $e = 1 : E$ **do**
14:             **for** $b = 1 : \mathcal{B}$ **do**
15:                 Compute prompt-generated feature $f_{g_l,e_l^t}(x)$ as in Eq.1 to 3
16:                 Compute logits with prototypes $f_{\phi_l}(f_{g_l,e_l^t}(x) \cup W^t)$
17:                 Compute loss $\mathcal{L}_{total} = \mathcal{L} + \mathcal{L}_{match}$ as in Eq.9
18:                 Update local parameters $(g_l, e_l^t, \phi_l)$ based on $\mathcal{L}_{total}$
19:             **end for**
20:             **if** $W_l^t = \emptyset$ **then**
21:                 Compute prototype $\mu_{lc}^t$, ${\sigma_{lc}^t}^2$ for class-$c$ available in $\mathcal{T}_l^{r,t}$ Eq.6 and 7
22:             **end if**
23:             Augment the prototypes Eq. 8
24:             Unify mean and augmented prototype into $W_l^t$
25:         **end for**
26:         Compute prototype $\mu_{lc}^t$, ${\sigma_{lc}^t}^2$ for class-$c$ available in $\mathcal{T}_l^{r,t}$ Eq.6 and 7
27:         Set $W_l^t = \{(\mu_{lc}^t, {\sigma_{lc}^t}^2)\}$, for class-$c$ available in $\mathcal{T}_l^{r,t}$
28:         Store local parameters $(g_l, e_l^t, \phi_l, W_l^t)$
29:         Compute clients' weight $\omega_l^t$
30:         Send local parameters $(g_l, e_l^t, \phi_l, W_l^t)$ and weight $\omega_l^t$ to server
31:         **Server executes**:
32:         Receives selected local clients $S_l$ parameters $(g_l, e_l^t, \phi_l, W_l^t)$
33:         Aggregates (using weighted aggregation) clients' prompt, and head layer into global parameters and weight $\omega_l^t$ $(g_G, e_G^t, \phi_G) \leftarrow (g_l, e_l^t, \phi_l)$ as in Eq10
34:         Aggregates (using weighted aggregation) clients' prototypes into global prototypes $W_G^t \leftarrow W_l^t$ as in Eq11
35:         Send global parameters $(g_G, e_G^t, \phi_G)$ to clients for the next round
36:     **end for**
37: **end for**
38: **Output:** Global parameters $(g_G, e_G^t, \phi_G)$ and local parameters $(g_l, e_l^t, \phi_l), l \in 1..N$

---

| Method | 10 | 20 | 30 | 40 | 50 | 60 | 70 | 80 | 90 | 100 | Avg | PD | Imp |
|---|---|---|---|---|---|---|---|---|---|---|---|---|---|
| iCaRL+FL | 89.00 | 55.00 | 57.00 | 52.30 | 50.30 | 49.30 | 46.30 | 41.70 | 40.30 | 36.70 | 51.79 | 52.30 | 36.21 |
| BiC+FL | 88.70 | 63.30 | 61.30 | 56.70 | 53.00 | 51.70 | 48.00 | 44.00 | 42.70 | 40.70 | 55.01 | 48.00 | 32.99 |
| PODNet+FL | 89.00 | 71.30 | 69.00 | 63.30 | 59.00 | 55.30 | 50.70 | 48.70 | 45.30 | 45.00 | 59.66 | 44.00 | 28.34 |
| DDE+iCaRL+FL | 88.00 | 70.00 | 67.30 | 62.00 | 57.30 | 54.70 | 50.30 | 48.30 | 45.70 | 44.30 | 58.79 | 43.70 | 29.21 |
| GeoDL+iCaRL+FL | 87.00 | 76.00 | 70.30 | 64.30 | 60.70 | 57.30 | 54.70 | 50.30 | 48.30 | 46.30 | 61.52 | 40.70 | 26.48 |
| SS-IL+FL | 88.30 | 66.30 | 54.00 | 54.00 | 44.70 | 54.70 | 50.00 | 47.70 | 45.30 | 44.00 | 54.90 | 44.30 | 33.10 |
| DyTox+FL | 86.20 | 76.90 | 73.30 | 69.50 | 62.10 | 62.70 | 58.10 | 57.20 | 55.40 | 52.10 | 65.35 | 34.10 | 22.65 |
| AFC+FL | 85.60 | 73.00 | 65.10 | 62.40 | 54.00 | 53.10 | 51.90 | 47.00 | 46.10 | 43.60 | 58.18 | 42.00 | 29.82 |
| GLFC | 90.00 | 82.30 | 77.00 | 72.30 | 65.00 | 66.30 | 59.70 | 56.30 | 50.30 | 50.00 | 66.92 | 40.00 | 21.08 |
| LGA | 89.60 | 83.20 | 79.30 | 76.10 | 72.90 | 71.70 | 68.40 | 65.70 | 64.70 | 62.90 | 73.45 | 26.70 | 14.55 |
| Fed-DualPrompt | 96.80 | 81.55 | 75.72 | 72.56 | 69.54 | 67.34 | 69.93 | 69.18 | 69.90 | 69.74 | 74.23 | 27.06 | 13.77 |
| **PIP(Ours)** | **98.70** | **92.85** | **89.42** | **87.58** | **87.01** | **85.32** | **85.19** | **84.79** | **84.84** | **84.31** | **88.00** | **14.39** | **-** |

Table 4: Complete numerical result of consolidated methods on CIFAR-100 Dataset. "Avg" denotes the average accuracy of all tasks, "PD" denotes performance drop, and "Imp" denotes improvement/gap of PIP compared to the respective method

| Method | 10 | 20 | 30 | 40 | 50 | 60 | 70 | 80 | 90 | 100 | Avg | PD | Imp |
|---|---|---|---|---|---|---|---|---|---|---|---|---|---|
| iCaRL+FL | 74.00 | 62.30 | 56.30 | 47.70 | 46.00 | 40.30 | 37.70 | 34.30 | 33.30 | 32.70 | 46.46 | 41.30 | 44.56 |
| BiC+FL | 74.30 | 63.00 | 57.70 | 51.30 | 48.30 | 46.00 | 42.70 | 37.70 | 35.30 | 34.00 | 49.03 | 40.30 | 41.99 |
| PODNet+FL | 74.30 | 64.00 | 59.00 | 56.70 | 52.70 | 50.30 | 47.00 | 43.30 | 40.00 | 38.30 | 52.56 | 36.00 | 38.46 |
| DDE+iCaRL+FL | 76.00 | 57.70 | 58.00 | 56.30 | 53.30 | 50.70 | 47.30 | 44.00 | 40.70 | 39.00 | 52.30 | 37.00 | 38.72 |
| GeoDL+iCaRL+FL | 74.00 | 63.30 | 54.70 | 53.30 | 50.70 | 46.70 | 41.30 | 39.70 | 38.30 | 37.00 | 49.90 | 37.00 | 41.12 |
| SS-IL+FL | 69.70 | 60.00 | 50.30 | 45.70 | 41.70 | 44.30 | 39.00 | 38.30 | 38.00 | 37.30 | 46.43 | 32.40 | 44.59 |
| DyTox+FL | 76.30 | 68.30 | 64.80 | 58.60 | 45.40 | 41.30 | 39.70 | 37.10 | 36.20 | 35.30 | 50.30 | 41.00 | 40.72 |
| AFC+FL | 82.50 | 74.10 | 66.80 | 60.00 | 48.00 | 44.30 | 42.50 | 40.90 | 39.00 | 36.10 | 53.42 | 46.40 | 37.60 |
| GLFC | 73.00 | 69.30 | 68.00 | 61.00 | 58.30 | 54.00 | 51.30 | 48.00 | 44.30 | 42.70 | 56.99 | 30.30 | 34.03 |
| LGA | 83.00 | 74.20 | 72.30 | 72.20 | 68.10 | 65.80 | 64.00 | 59.60 | 58.40 | 57.50 | 67.51 | 25.50 | 23.51 |
| Fed-DualPrompt | 97.57 | 83.55 | 82.52 | 79.22 | 76.64 | 76.19 | 75.82 | 75.85 | 75.56 | 78.50 | 80.14 | 19.07 | 10.88 |
| **PIP(Ours)** | **98.43** | **90.92** | **90.42** | **90.68** | **90.17** | **90.68** | **90.26** | **90.44** | **88.94** | **89.21** | **91.02** | **9.23** | **-** |

Table 5: Complete numerical result of consolidated methods on MiniImageNet Dataset. "Avg" denotes the average accuracy of all tasks, "PD" denotes performance drop, and "Imp" denotes improvement/gap of PIP compared to the respective method

| Method | 20 | 40 | 60 | 80 | 100 | 120 | 140 | 160 | 180 | 200 | Avg | PD | Imp |
|---|---|---|---|---|---|---|---|---|---|---|---|---|---|
| iCaRL+FL | 63.00 | 53.00 | 48.00 | 41.70 | 38.00 | 36.00 | 33.30 | 30.70 | 29.70 | 28.00 | 40.14 | 35.00 | 46.12 |
| BiC+FL | 65.30 | 52.70 | 49.30 | 46.00 | 40.30 | 38.30 | 35.70 | 33.00 | 31.70 | 29.00 | 42.13 | 36.30 | 44.13 |
| PODNet+FL | 66.70 | 53.30 | 50.00 | 47.30 | 43.70 | 42.70 | 40.00 | 37.30 | 33.70 | 31.30 | 44.60 | 35.40 | 41.66 |
| DDE+iCaRL+FL | 69.00 | 52.00 | 50.70 | 47.00 | 43.30 | 42.00 | 39.30 | 37.00 | 33.00 | 31.30 | 44.46 | 37.70 | 41.80 |
| GeoDL+iCaRL+FL | 66.30 | 54.30 | 52.00 | 48.70 | 45.00 | 42.00 | 39.30 | 36.00 | 32.70 | 30.00 | 44.63 | 36.30 | 41.63 |
| SS-IL+FL | 62.00 | 48.70 | 40.00 | 38.00 | 37.00 | 35.00 | 32.30 | 30.30 | 28.70 | 27.00 | 37.90 | 35.00 | 48.36 |
| DyTox+FL | 73.20 | 66.60 | 48.00 | 47.10 | 41.60 | 40.80 | 37.40 | 36.20 | 32.80 | 30.60 | 45.43 | 42.60 | 40.83 |
| AFC+FL | 73.70 | 59.10 | 50.80 | 43.10 | 37.00 | 35.20 | 32.60 | 32.00 | 28.90 | 27.10 | 41.95 | 46.60 | 44.31 |
| GLFC | 66.00 | 58.30 | 55.30 | 51.00 | 47.70 | 45.30 | 43.00 | 40.00 | 37.30 | 35.00 | 47.89 | 31.00 | 38.37 |
| LGA | 70.30 | 64.00 | 60.30 | 58.00 | 55.80 | 53.10 | 47.90 | 45.30 | 39.80 | 37.30 | 53.18 | 33.00 | 33.08 |
| Fed-DualPrompt | 86.27 | 74.55 | 71.16 | 65.88 | 63.33 | 62.03 | 61.34 | 59.88 | 58.20 | 61.29 | 66.39 | 24.98 | 19.87 |
| **PIP(Ours)** | **92.77** | **86.35** | **86.62** | **87.53** | **86.73** | **87.02** | **85.29** | **84.92** | **83.95** | **81.40** | **86.26** | **11.36** | **-** |

Table 6: Complete numerical result of consolidated methods on TinyImageNet Dataset. "Avg" denotes the average accuracy of all tasks, "PD" denotes performance drop, and "Imp" denotes improvement/gap of PIP compared to the respective method

$$O(Baseline) = O(1) + T.R_T.(L.O(1client) + O(server)) \tag{27}$$
$$O(Baseline) = O(1) + T.R_T.(L.O(1client) + O(1)) \tag{28}$$

$$O(Baseline) = O(1) + T.R_T.L.O(E.\sum_{b=1}^{\beta} N_{bl}^t + O(1)) + O(T.R_T.) \tag{29}$$

$$O(Baseline) = O(1) + O(T.R_T.L.E.\sum_{b=1}^{\beta} N_{bl}^t) + O(T.R_T.L.) + O(T.R_T) \tag{30}$$

$$O(Baseline) = O(T.R_T.L.E.\sum_{b=1}^{\beta} N_{bl}^t) + O(T.R_T.L.) + O(T.R_T) \tag{31}$$

From the definition above that $\sum_{b=1}^{\beta} N_{bl}^t = N_l^t$, $R_T = R/T$, $N_l = T.N_l^t = T$, and $L \geq 1$ therefore the complexity of the baseline will be

$$O(Baseline) = O(T.R/T.L.E.N_l^t) + O(T.R/T.L.) + O(T.R/T) \tag{32}$$

$$O(Baseline) = O(R.L.E.N_l^t) + O(R.L) + O(R) \tag{33}$$

$$O(Baseline) = O(R.L.E.N_l^t) \tag{34}$$

Since $N_l^t < N_l$ and E is set as a small constant in our method i.e. 2, then the baseline complexity will be:

$$O(Baseline) = O(R.L.N_l) \tag{35}$$

**PIP Complexity:** Following the pseudo-code in Algorithm 2, PIP generates prototypes when its prototype set is empty after a local epoch of a round-$r$ (line 21), augment the prototypes in each local epoch (23), and updates the prototypes after local epochs (line 26-27). Knowing that generating prototypes from $\mathcal{T}_l^{r,t}$ costs $O(N_l^t)$, augmenting the prototypes costs $O(1)$ since it runs $m \in \{1..5\}$ times, then the PIP complexity will be:

$$O(PIP) = O(1) + T.R_T.(O(clients) + O(server)) \tag{36}$$
$$O(PIP) = O(1) + T.R_T.(L.O(1client) + O(server)) \tag{37}$$
$$O(PIP) = O(1) + T.R_T.(L.O(1client) + O(1)) \tag{38}$$

$$O(PIP) = O(1) + T.R_T.L.O(E.(\sum_{b=1}^{\beta} N_{bl}^t + N_l^t) + O(N_l^t) + O(1)) + O(T.R_T) \tag{39}$$

Since we have $\sum_{b=1}^{\beta} N_{bl}^t = N_l^t$, then we have

$$O(PIP) = O(1) + T.R_T.L.O(E.(N_l^t + N_l^t) + O(N_l^t) + O(1)) + O(T.R_T) \tag{40}$$

$$O(PIP) = O(1) + T.R_T.L.O(E.(N_l^t) + O(N_l^t) + O(1)) + O(T.R_T) \tag{41}$$

$$O(PIP) = O(1) + O(T.R_T.L.E.N_l^t) + O(T.R_T.L.N_l^t) + O(T.R_T.L) + O(T.R_T) \tag{42}$$

$$O(PIP) = O(T.R_T.L.E.N_l^t) + O(T.R_T.L.N_l^t) + O(T.R_T.L) + O(T.R_T) \tag{43}$$

Substituting the equalities in the previous definition that $R_T = R/T$ and $N_l = T.N_l^t = T$ the complexity of PIP will be

$$O(PIP) = O(T.R/T.L.E.N_l^t) + O(T.R/T.L.N_l^t) + O(T.R/T.L) + O(T.R/T) \tag{44}$$

$$O(PIP) = O(R.L.E.N_l^t) + O(R.L.N_l^t) + O(R.L) + O(R) \tag{45}$$

$$O(Baseline) = O(R.L.E.N_l^t) \tag{46}$$

Since $N_l^t < N_l$ and E is set as a small constant in our method i.e. 2, then the PIP complexity will be:

$$O(Baseline) = O(R.L.N_l) \tag{47}$$

Our derivation shows that the baseline and our proposed method (PIP) have the same complexity i.e. $O(R.L.N_l)$ where $R$ is total global rounds, $L$ is the number of selected local clients in each round and $N_l$ is the number of samples in each client.

| Config. | Prompt | Proto | W.Agg | Head | 10 | 20 | 30 | 40 | 50 | Avg | PD | Imp |
|---|---|---|---|---|---|---|---|---|---|---|---|---|
| A | v | | | | 82.15 | 69.97 | 67.06 | 64.52 | 63.78 | - | - | - |
| B | v | v | | | 90.99 | 85.46 | 83.65 | 81.99 | 80.90 | - | - | - |
| C | v | | v | | 82.13 | 69.85 | 67.18 | 65.15 | 64.11 | - | - | - |
| D | v | v | v | | 91.37 | 86.46 | 84.37 | 82.79 | 81.87 | - | - | - |
| E | v | | | v | 96.60 | 83.20 | 77.10 | 74.08 | 70.86 | - | - | - |
| F | v | v | | v | 98.90 | 91.80 | 88.80 | 87.25 | 86.78 | - | - | - |
| G | v | | v | v | 98.60 | 71.80 | 68.73 | 67.68 | 68.44 | - | - | - |
| PIP | v | v | v | v | **98.60** | **92.90** | **89.57** | **87.78** | **87.14** | - | - | - |
| Config. | Prompt | Proto | W.Agg | Head | 60 | 70 | 80 | 90 | 100 | Avg | PD | Imp |
| A | v | | | | 62.99 | 62.72 | 61.90 | 61.12 | 60.66 | 65.69 | 21.49 | 22.59 |
| B | v | v | | | 79.07 | 78.63 | 78.99 | 78.69 | 77.56 | 81.59 | **13.43** | 6.69 |
| C | v | | v | | 63.50 | 62.84 | 62.78 | 62.16 | 61.35 | 66.10 | 20.78 | 22.18 |
| D | v | v | v | | 79.72 | 79.78 | 79.74 | 79.54 | 78.47 | 82.41 | **12.90** | 5.87 |
| E | v | | | v | 68.73 | 71.83 | 70.00 | 70.71 | 70.68 | 75.38 | 25.92 | 12.90 |
| F | v | v | | v | 85.52 | 85.29 | 84.83 | 84.86 | 84.17 | 87.82 | 14.73 | 0.46 |
| G | v | | v | v | 69.48 | 70.31 | 70.98 | 69.77 | 69.94 | 72.57 | 28.66 | 15.71 |
| PIP | v | v | v | v | **85.93** | **85.60** | **85.45** | **85.22** | **84.60** | **88.28** | 14.00 | - |

Table 7: Complete numerical result of ablation study on CIFAR100 dataset (T=10) on one seeded run i.e. 2021. "Avg" denotes the average accuracy of all tasks, "PD" denotes performance drop, and "Imp" denotes improvement/gap of PIP compared to the respective configuration

# F COMPLETE NUMERICAL RESULT OF ABLATION STUDY

# G COMPLETE NUMERICAL RESULT OF EXPERIMENT ON 3 BENCHMARK DATASETS T=5 AND T=20

In this section, we present the detailed numerical results on different task sizes (T=5) and (T=20) on CIFAR100, MiniImageNet, and TinyImageNet datasets.

| Method | CIFAR100 (T=5) | | | | | | MiniImageNet (T=5) | | | | | |
|---|---|---|---|---|---|---|---|---|---|---|---|---|
| | 20 | 40 | 60 | 80 | 100 | Avg. | 20 | 40 | 60 | 80 | 100 | Avg. |
| iCaRL+FL | 77.00 | 59.60 | 51.90 | 44.40 | 39.60 | 54.50 | 73.50 | 56.20 | 46.20 | 40.20 | 35.50 | 50.32 |
| BiC+FL | 78.40 | 60.40 | 53.20 | 47.50 | 41.20 | 56.14 | 72.60 | 56.80 | 49.20 | 43.50 | 38.70 | 52.16 |
| PODNet+FL | 77.60 | 62.10 | 56.30 | 50.80 | 43.30 | 58.02 | 73.10 | 58.40 | 53.20 | 46.50 | 43.40 | 54.92 |
| DDE+iCaRL+FL | 77.00 | 60.20 | 55.70 | 49.30 | 42.50 | 56.94 | 72.30 | 57.20 | 51.70 | 44.30 | 41.30 | 53.36 |
| GeoDL+iCaRL+FL | 72.50 | 61.10 | 54.00 | 49.50 | 44.50 | 56.32 | 71.80 | 59.60 | 52.30 | 46.10 | 42.50 | 54.46 |
| SS-IL+FL | 78.10 | 61.80 | 52.80 | 48.80 | 46.00 | 57.50 | 66.50 | 52.10 | 42.60 | 36.70 | 36.50 | 46.88 |
| DyTox+FL | 78.80 | 70.50 | 63.90 | 59.90 | 55.90 | 65.80 | 69.60 | 64.20 | 59.10 | 53.40 | 48.50 | 58.96 |
| AFC+FL | 71.10 | 63.80 | 58.40 | 53.60 | 46.40 | 58.66 | 78.00 | 64.50 | 57.00 | 51.30 | 47.30 | 59.62 |
| GLFC | 83.70 | 75.50 | 66.50 | 62.10 | 53.80 | 68.32 | 79.70 | 73.40 | 65.20 | 58.10 | 51.80 | 65.64 |
| LGA | 83.30 | 77.30 | 72.80 | 67.80 | 63.70 | 72.98 | 78.90 | 75.50 | 68.10 | 62.10 | 61.90 | 69.30 |
| Fed-DualPrompt | 93.15 | 76.88 | 67.55 | 66.56 | 66.06 | 74.04 | 94.35 | 81.13 | 74.67 | 73.51 | 76.67 | 80.06 |
| **PIP(Ours)** | **96.75** | **86.35** | **83.57** | **80.68** | **79.71** | **85.41** | **95.30** | **89.68** | **88.52** | **88.44** | **87.32** | **89.85** |

Table 8: Complete numerical results on CIFAR100 and MniImageNet dataset (T=5) in one-seeded run i.e. 2021

# H COMPLETE NUMERICAL RESULT OF EXPERIMENT ON SMALLER LOCAL CLIENTS

In this section, we present the detailed numerical results of smaller local clients on CIFAR100, and TinyImageNet datasets.

| Method | 40 | 80 | 120 | 160 | 200 | Avg. |
|---|---|---|---|---|---|---|
| iCaRL+FL | 65.00 | 48.00 | 42.70 | 38.70 | 35.00 | 45.88 |
| BiC+FL | 65.70 | 48.70 | 43.00 | 40.30 | 35.70 | 46.68 |
| PODNet+FL | 66.00 | 50.30 | 44.70 | 41.30 | 37.00 | 47.86 |
| DDE+iCaRL+FL | 63.00 | 51.30 | 45.30 | 41.00 | 36.00 | 47.32 |
| GeoDL+iCaRL+FL | 65.30 | 50.00 | 45.00 | 40.70 | 36.00 | 47.4 |
| SS-IL+FL | 65.00 | 42.30 | 38.30 | 35.00 | 30.30 | 42.18 |
| DyTox+FL | 58.60 | 43.10 | 41.60 | 37.20 | 32.90 | 42.68 |
| AFC+FL | 62.50 | 52.10 | 45.70 | 43.20 | 35.70 | 47.84 |
| GLFC | 66.00 | 55.30 | 49.00 | 45.00 | 40.30 | 51.12 |
| LGA | 67.70 | 59.80 | 53.50 | 47.90 | 43.80 | 54.54 |
| Fed-DualPrompt | 85.10 | 67.23 | 60.25 | 55.99 | 58.59 | 65.43 |
| **PIP(Ours)** | **91.20** | **84.95** | **83.80** | **81.86** | **79.16** | **84.19** |

Table 9: Complete numerical results on TinyImageNet dataset (T=5) in one seeded run i.e. 2021

| Method | 5 | 10 | 15 | 20 | 25 | 30 | 35 | 40 | 45 | 50 | Avg |
|---|---|---|---|---|---|---|---|---|---|---|---|
| iCaRL+FL | 82.00 | 80.00 | 67.00 | 62.00 | 61.30 | 60.30 | 57.00 | 54.30 | 53.00 | 51.70 | - |
| BiC+FL | 82.00 | 77.30 | 68.30 | 64.00 | 63.70 | 62.30 | 60.30 | 58.70 | 55.00 | 53.30 | - |
| PODNet+FL | 83.00 | 76.30 | 70.30 | 68.00 | 66.30 | 67.00 | 65.30 | 61.70 | 61.30 | 58.70 | - |
| DDE+iCaRL+FL | 83.00 | 75.30 | 69.70 | 65.00 | 67.00 | 63.70 | 59.30 | 58.00 | 60.00 | 55.30 | - |
| GeoDL+iCaRL+FL | 82.00 | 78.30 | 71.30 | 67.70 | 68.00 | 65.30 | 64.30 | 60.00 | 58.70 | 56.00 | - |
| SS-IL+FL | 83.00 | 73.30 | 63.70 | 61.30 | 60.30 | 59.30 | 57.30 | 56.00 | 54.70 | 53.30 | - |
| DyTox+FL | 79.60 | 78.30 | 67.10 | 65.60 | 68.50 | 64.30 | 63.70 | 61.00 | 58.80 | 59.00 | - |
| AFC+FL | 75.60 | 69.60 | 57.10 | 58.50 | 45.50 | 55.40 | 51.40 | 50.40 | 45.20 | 42.40 | - |
| GLFC | 82.20 | 82.50 | 74.90 | 75.20 | 73.30 | 71.50 | 70.10 | 67.70 | 64.60 | 65.90 | - |
| LGA | 85.80 | 85.90 | 80.70 | 78.90 | 78.40 | 74.60 | 75.10 | 71.30 | 68.90 | 69.20 | - |
| Fed-DualPrompt | 94.80 | 91.40 | 85.13 | 85.00 | 84.20 | 80.17 | 79.14 | 76.75 | 76.60 | 75.46 | - |
| **PIP(Ours)** | **98.40** | **96.70** | **92.67** | **91.65** | **90.88** | **88.47** | **88.37** | **86.87** | **87.00** | **86.30** | - |
| Method | 55 | 60 | 65 | 70 | 75 | 80 | 85 | 90 | 95 | 100 | Avg |
| iCaRL+FL | 50.30 | 50.00 | 48.70 | 48.00 | 46.70 | 45.00 | 45.00 | 44.00 | 43.30 | 42.70 | 46.37 |
| BiC+FL | 52.00 | 51.30 | 50.30 | 49.70 | 48.00 | 47.00 | 46.30 | 45.70 | 45.30 | 44.30 | 63.23 |
| PODNet+FL | 56.30 | 55.00 | 54.00 | 53.00 | 51.00 | 50.30 | 49.30 | 48.00 | 48.30 | 47.70 | 44.82 |
| DDE+iCaRL+FL | 54.70 | 54.00 | 53.30 | 52.00 | 50.70 | 50.00 | 49.30 | 48.70 | 48.00 | 47.30 | 45.39 |
| GeoDL+iCaRL+FL | 55.30 | 55.00 | 53.70 | 53.00 | 51.70 | 50.70 | 50.00 | 49.00 | 49.30 | 48.00 | 47.67 |
| SS-IL+FL | 52.30 | 52.00 | 51.30 | 50.70 | 50.00 | 49.30 | 49.00 | 48.30 | 48.00 | 47.70 | 46.88 |
| DyTox+FL | 56.20 | 58.50 | 58.30 | 58.20 | 55.00 | 51.80 | 49.70 | 48.70 | 49.00 | 52.70 | 49.00 |
| AFC+FL | 41.30 | 35.60 | 37.10 | 37.80 | 38.90 | 35.20 | 34.40 | 34.50 | 36.20 | 33.80 | 38.34 |
| GLFC | 63.70 | 64.20 | 62.00 | 61.00 | 60.20 | 58.90 | 57.60 | 59.30 | 56.80 | 56.80 | 46.29 |
| LGA | 68.30 | 67.70 | 65.50 | 65.60 | 64.00 | 63.00 | 63.10 | 63.70 | 61.60 | 60.50 | 47.62 |
| Fed-DualPrompt | 75.60 | 72.80 | 71.92 | 72.37 | 70.89 | 70.94 | 71.16 | 71.11 | 70.89 | 70.30 | 60.58 |
| **PIP(Ours)** | **84.53** | **83.70** | **82.88** | **83.16** | **82.08** | **82.54** | **82.69** | **82.39** | **82.78** | **81.49** | **70.27** |

Table 10: Complete numerical results on CIFAR100 dataset (T=20) in one-seeded run i.e. 2021

# I    COMPLETE NUMERICAL RESULT OF EXPERIMENT ON SMALLER GLOBAL ROUNDS

In this section, we present the detailed numerical results of smaller global rounds on CIFAR100, and TinyImageNet datasets.

| Method | 5 | 10 | 15 | 20 | 25 | 30 | 35 | 40 | 45 | 50 | Avg |
|---|---|---|---|---|---|---|---|---|---|---|---|
| iCaRL+FL | 83.00 | 66.00 | 61.30 | 56.00 | 56.30 | 53.00 | 49.70 | 47.00 | 46.30 | 46.00 | - |
| BiC+FL | 82.30 | 64.70 | 59.00 | 58.30 | 57.00 | 54.70 | 52.30 | 50.30 | 49.00 | 47.70 | - |
| PODNet+FL | 81.70 | 63.30 | 60.30 | 59.30 | 58.30 | 56.30 | 55.00 | 53.30 | 51.70 | 50.00 | - |
| DDE+iCaRL+FL | 80.00 | 60.70 | 58.70 | 56.30 | 57.00 | 55.30 | 53.00 | 51.70 | 50.30 | 49.30 | - |
| GeoDL+iCaRL+FL | 82.30 | 66.30 | 62.70 | 61.00 | 60.30 | 58.00 | 56.30 | 55.30 | 53.00 | 51.30 | - |
| SS-IL+FL | 80.00 | 65.30 | 61.70 | 57.30 | 56.30 | 54.00 | 51.30 | 50.00 | 49.30 | 48.30 | - |
| DyTox+FL | 71.60 | 52.70 | 61.60 | 53.20 | 56.80 | 48.90 | 45.70 | 49.40 | 39.10 | 44.10 | - |
| AFC+FL | 72.40 | 53.00 | 51.80 | 38.10 | 41.40 | 39.60 | 41.20 | 37.20 | 33.70 | 32.40 | - |
| GLFC | 84.00 | 71.70 | 70.00 | 69.30 | 67.30 | 66.30 | 61.00 | 60.70 | 59.30 | 58.70 | - |
| LGA | 78.80 | 79.40 | 76.80 | 73.50 | 69.80 | 68.50 | 67.30 | 66.10 | 63.80 | 62.10 | - |
| Fed-DualPrompt | 96.20 | 94.50 | 88.73 | 86.80 | 86.04 | 86.80 | 85.83 | 85.63 | 86.11 | 85.90 | - |
| **PIP(Ours)** | **97.80** | **96.70** | **94.33** | **92.45** | **91.92** | **91.70** | **91.09** | **90.80** | **91.07** | **90.34** | - |
| Method | 55 | 60 | 65 | 70 | 75 | 80 | 85 | 90 | 95 | 100 | Avg |
| iCaRL+FL | 44.00 | 42.30 | 40.00 | 39.70 | 37.30 | 36.00 | 34.70 | 34.30 | 33.00 | 32.00 | 37.33 |
| BiC+FL | 46.70 | 44.00 | 42.70 | 41.30 | 40.30 | 38.00 | 37.00 | 36.30 | 34.70 | 33.00 | 39.40 |
| PODNet+FL | 49.30 | 48.00 | 47.00 | 45.30 | 44.70 | 43.70 | 42.00 | 39.70 | 38.70 | 37.00 | 43.54 |
| DDE+iCaRL+FL | 48.70 | 48.30 | 47.70 | 46.70 | 45.70 | 44.30 | 42.30 | 40.00 | 38.30 | 37.30 | 43.93 |
| GeoDL+iCaRL+FL | 50.00 | 48.70 | 48.00 | 46.30 | 45.00 | 44.00 | 41.70 | 40.00 | 38.00 | 36.70 | 43.84 |
| SS-IL+FL | 47.00 | 45.00 | 44.30 | 43.00 | 41.30 | 40.70 | 39.30 | 38.70 | 37.00 | 36.00 | 41.23 |
| DyTox+FL | 37.70 | 35.20 | 33.60 | 31.50 | 28.60 | 27.30 | 27.10 | 26.50 | 25.80 | 24.90 | 29.82 |
| AFC+FL | 29.70 | 33.50 | 29.60 | 30.20 | 25.10 | 25.10 | 26.10 | 24.60 | 24.00 | 23.50 | 27.14 |
| GLFC | 55.30 | 53.00 | 52.00 | 50.30 | 49.70 | 47.30 | 46.00 | 42.70 | 40.30 | 39.00 | 47.56 |
| LGA | 60.60 | 59.80 | 57.20 | 56.80 | 55.10 | 54.70 | 54.10 | 53.20 | 51.60 | 48.20 | 55.13 |
| Fed-DualPrompt | 85.82 | 85.97 | 85.78 | 84.63 | 84.71 | 84.53 | 82.81 | 81.73 | 81.96 | 82.35 | 84.03 |
| **PIP(Ours)** | **89.69** | **90.03** | **89.97** | **89.43** | **89.20** | **89.14** | **88.38** | **87.39** | **87.89** | **88.04** | **88.92** |

Table 11: Complete numerical results on MiniImageNet dataset (T=20) in one-seeded run i.e. 2021

| Method | 10 | 20 | 30 | 40 | 50 | 60 | 70 | 80 | 90 | 100 | Avg |
|---|---|---|---|---|---|---|---|---|---|---|---|
| iCaRL+FL | 67.00 | 59.30 | 54.00 | 48.30 | 46.70 | 44.70 | 43.30 | 39.00 | 37.30 | 33.00 | - |
| BiC+FL | 67.30 | 59.70 | 54.70 | 50.00 | 48.30 | 45.30 | 43.00 | 40.70 | 38.00 | 33.70 | - |
| PODNet+FL | 69.00 | 59.30 | 55.00 | 51.70 | 50.00 | 46.70 | 43.70 | 41.00 | 39.30 | 38.00 | - |
| DDE+iCaRL+FL | 70.00 | 59.30 | 53.30 | 51.00 | 48.30 | 45.70 | 42.30 | 40.00 | 38.00 | 36.30 | - |
| GeoDL+iCaRL+FL | 66.30 | 56.70 | 51.00 | 49.70 | 44.70 | 42.30 | 41.00 | 39.00 | 37.30 | 35.00 | - |
| SS-IL+FL | 66.70 | 54.00 | 47.70 | 45.30 | 42.30 | 42.00 | 40.70 | 38.00 | 36.00 | 34.30 | - |
| DyTox+FL | 77.60 | 70.20 | 63.40 | 56.60 | 52.00 | 44.60 | 51.60 | 39.60 | 41.50 | 39.00 | - |
| AFC+FL | 74.00 | 62.90 | 57.60 | 54.20 | 45.10 | 44.40 | 40.70 | 36.90 | 33.00 | 33.60 | - |
| GLFC | 68.70 | 63.30 | 61.70 | 57.30 | 56.00 | 53.00 | 50.30 | 47.70 | 46.30 | 45.00 | - |
| LGA | 74.00 | 67.60 | 64.90 | 61.00 | 58.90 | 55.70 | 53.60 | 51.30 | 50.10 | 48.80 | - |
| Fed-DualPrompt | 79.40 | 78.90 | 79.00 | 78.10 | 76.48 | 77.53 | 77.14 | 76.85 | 76.04 | 75.48 | - |
| **PIP(Ours)** | **89.20** | **87.80** | **86.93** | **85.00** | **84.12** | **85.70** | **86.31** | **86.62** | **86.76** | **86.44** | - |
| Method | 110 | 120 | 130 | 140 | 150 | 160 | 170 | 180 | 190 | 200 | Avg |
| iCaRL+FL | 32.00 | 30.30 | 28.00 | 27.00 | 26.30 | 25.30 | 24.70 | 24.00 | 22.70 | 22.00 | 36.75 |
| BiC+FL | 32.70 | 32.30 | 30.30 | 29.00 | 27.70 | 27.30 | 26.00 | 25.70 | 24.30 | 23.30 | 37.97 |
| PODNet+FL | 37.00 | 35.70 | 34.70 | 34.00 | 33.00 | 32.30 | 31.00 | 30.00 | 29.30 | 28.00 | 40.94 |
| DDE+iCaRL+FL | 35.00 | 33.70 | 32.00 | 31.00 | 30.30 | 30.00 | 28.70 | 28.30 | 27.30 | 26.00 | 39.33 |
| GeoDL+iCaRL+FL | 33.70 | 32.00 | 31.00 | 30.30 | 28.70 | 28.00 | 27.30 | 26.30 | 25.00 | 24.70 | 37.50 |
| SS-IL+FL | 33.00 | 31.00 | 29.30 | 28.30 | 27.70 | 27.00 | 26.30 | 26.00 | 25.00 | 24.30 | 36.25 |
| DyTox+FL | 37.80 | 31.20 | 34.20 | 30.60 | 29.80 | 29.20 | 28.30 | 27.50 | 26.80 | 15.30 | 41.34 |
| AFC+FL | 30.80 | 28.90 | 27.10 | 22.80 | 24.50 | 23.60 | 22.10 | 20.70 | 18.40 | 18.10 | 35.97 |
| GLFC | 42.70 | 41.00 | 40.00 | 39.30 | 38.00 | 36.70 | 35.30 | 34.00 | 33.00 | 31.70 | 46.05 |
| LGA | 45.20 | 43.70 | 42.80 | 41.20 | 40.50 | 38.90 | 37.40 | 36.60 | 35.10 | 33.80 | 49.06 |
| Fed-DualPrompt | 75.20 | 75.37 | 75.23 | 73.90 | 73.76 | 72.80 | 73.27 | 72.20 | 72.28 | 71.69 | 75.53 |
| **PIP(Ours)** | **86.80** | **86.22** | **86.22** | **85.67** | **85.81** | **84.30** | **84.35** | **83.49** | **83.72** | **83.33** | **85.74** |

Table 12: Complete numerical results on TinyImageNet dataset (T=20) in one-seeded run i.e. 2021

| Method | 10 | 20 | 30 | 40 | 50 | 60 | 70 | 80 | 90 | 100 | Avg |
|---|---|---|---|---|---|---|---|---|---|---|---|
| GLFC-10clients | 90.00 | 82.30 | 77.00 | 72.30 | 65.00 | 66.30 | 59.70 | 56.30 | 50.30 | 50.00 | 66.92 |
| GLFC-3clients | 86.70 | 74.50 | 71.70 | 65.63 | 65.26 | 60.37 | 54.51 | 52.63 | 48.30 | 44.56 | 62.42 |
| GLFC-2clients | 85.00 | 71.80 | 65.10 | 64.58 | 54.96 | 51.00 | 44.79 | 49.69 | 41.42 | 45.61 | 57.39 |
| LGA-10clients | 89.60 | 83.20 | 79.30 | 76.10 | 72.90 | 71.70 | 68.40 | 65.70 | 64.70 | 62.90 | 73.45 |
| LGA-3clients | 85.50 | 74.30 | 75.70 | 72.80 | 71.02 | 66.50 | 64.57 | 61.04 | 61.22 | 56.05 | 68.87 |
| LGA-2clients | 86.40 | 76.35 | 78.07 | 67.75 | 63.40 | 62.67 | 61.57 | 56.61 | 60.39 | 53.61 | 66.68 |
| Fed-DualPrompt-10clients | 96.60 | 83.20 | 77.10 | 74.08 | 70.86 | 68.73 | 71.83 | 70.00 | 70.71 | 70.68 | 75.38 |
| Fed-DualPrompt-3clients | 88.30 | 83.20 | 80.27 | 77.45 | 74.26 | 71.73 | 73.91 | 73.34 | 72.42 | 74.72 | 76.96 |
| Fed-DualPrompt-2clients | 90.10 | 76.95 | 71.93 | 71.40 | 68.14 | 69.55 | 69.14 | 67.66 | 67.93 | 68.92 | 72.17 |
| PIP(Ours)-10clients | **98.60** | **92.90** | **89.57** | **87.78** | **87.14** | **85.93** | **85.60** | **85.45** | **85.22** | **84.60** | **88.28** |
| PIP(Ours)-3clients | **98.40** | **92.00** | **90.57** | **89.03** | **87.72** | **83.30** | **83.69** | **82.71** | **82.36** | **80.88** | **87.06** |
| PIP(Ours)-2clients | **96.70** | **90.15** | **88.80** | **86.70** | **83.34** | **79.20** | **79.34** | **78.30** | **77.94** | **77.28** | **83.78** |

Table 13: Complete numerical results on CIFAR100 (T=10) with smaller participating clients in one-seeded run i.e. 2021

| Method | 20 | 40 | 60 | 80 | 100 | 120 | 140 | 160 | 180 | 200 | Avg |
|---|---|---|---|---|---|---|---|---|---|---|---|
| GLFC-10clients | 66.00 | 58.30 | 55.30 | 51.00 | 47.70 | 45.30 | 43.00 | 40.00 | 37.30 | 35.00 | 47.89 |
| GLFC-3clients | 55.60 | 37.70 | 26.37 | 19.88 | - | - | - | - | - | - | 34.89 |
| GLFC-2clients | 53.40 | 39.20 | 27.40 | 11.08 | - | - | - | - | - | - | 32.77 |
| LGA-10clients | 70.30 | 64.00 | 60.30 | 58.00 | 55.80 | 53.10 | 47.90 | 45.30 | 39.80 | 37.30 | 53.18 |
| LGA-3clients | 73.30 | 69.55 | 60.30 | 52.75 | 47.16 | - | - | - | - | - | 60.61 |
| LGA-2clients | 71.4 | 59.75 | 57.167 | 54.425 | 49.96 | - | - | - | - | - | 58.54 |
| Fed-DualPrompt-10clients | 86.60 | 75.70 | 71.33 | 66.33 | 64.28 | 62.83 | 62.27 | 61.19 | 59.99 | 62.60 | 67.31 |
| Fed-DualPrompt-3clients | 73.00 | 70.15 | 68.30 | 65.20 | 63.78 | 62.22 | 63.23 | 62.44 | 64.47 | 68.35 | 66.11 |
| Fed-DualPrompt-2clients | 67.50 | 63.95 | 65.50 | 66.43 | 68.30 | 68.90 | 67.66 | 67.25 | 65.37 | 69.43 | 67.03 |
| PIP(Ours)-10clients | **92.70** | **87.90** | **86.83** | **88.23** | **87.04** | **87.18** | **85.96** | **85.13** | **84.62** | **82.35** | **86.79** |
| PIP(Ours)-3clients | **90.40** | **87.10** | **85.03** | **86.90** | **85.92** | **85.05** | **83.64** | **82.64** | **82.40** | **80.95** | **85.00** |
| PIP(Ours)-2clients | **90.10** | **84.40** | **82.97** | **84.23** | **82.46** | **82.33** | **80.33** | **78.51** | **78.61** | **77.34** | **82.13** |

Table 14: Complete numerical results on TinyImageNet (T=10) with smaller participating clients in one-seeded run i.e. 2021. GLFC and LGA performance is averaged from their first 4 and 5 tasks respectively due to crash

| Rounds | Method | 10 | 20 | 30 | 40 | 50 | 60 | 70 | 80 | 90 | 100 | **Avg** |
|---|---|---|---|---|---|---|---|---|---|---|---|---|
| 10 | Fed-DualPrompt | 88.30 | 83.20 | 80.27 | 77.45 | 74.26 | 71.73 | 73.91 | 73.34 | 72.42 | 74.72 | 76.96 |
| 10 | **PIP (Our)** | **98.40** | **92.00** | **90.57** | **89.03** | **87.72** | **83.30** | **83.69** | **82.71** | **82.36** | **80.88** | **87.06** |
| 8 | Fed-DualPrompt | 90.10 | 76.95 | 71.93 | 71.40 | 68.14 | 69.55 | 69.14 | 67.66 | 67.93 | 68.92 | 72.17 |
| 8 | **PIP (Our)** | **96.70** | **90.15** | **88.80** | **86.70** | **83.34** | **79.20** | **79.34** | **78.30** | **77.94** | **77.28** | **83.78** |
| 6 | Fed-DualPrompt | 85.90 | 85.40 | 81.37 | 78.53 | 74.14 | 72.68 | 73.16 | 72.63 | 73.77 | 73.78 | 77.13 |
| 6 | **PIP (Our)** | **97.60** | **91.40** | **87.03** | **86.10** | **85.36** | **82.13** | **82.00** | **81.69** | **80.80** | **80.15** | **85.43** |
| 4 | Fed-DualPrompt | 85.30 | 84.40 | 81.10 | 78.13 | 73.82 | 71.95 | 73.00 | 71.28 | 72.42 | 71.47 | 76.29 |
| 4 | **PIP (Our)** | **97.70** | **93.65** | **88.13** | **86.38** | **84.20** | **82.97** | **82.26** | **81.36** | **81.27** | **79.87** | **85.78** |
| 2 | Fed-DualPrompt | 88.20 | 81.45 | 78.07 | 76.45 | 74.10 | 72.60 | 71.06 | 69.98 | 69.34 | 68.91 | 75.02 |
| 2 | **PIP (Our)** | **97.80** | **91.05** | **86.40** | **83.33** | **83.42** | **81.27** | **81.13** | **82.01** | **81.01** | **80.28** | **84.77** |

Table 15: Complete numerical results on CIFAR100 (T=10, local clients = 3) with smaller rounds in one-seeded run i.e. 2021

| Rounds | Method | 20 | 40 | 60 | 80 | 100 | 120 | 140 | 160 | 180 | 200 | **Avg** |
|---|---|---|---|---|---|---|---|---|---|---|---|---|
| 10 | Fed-DualPrompt | 86.60 | 75.70 | 71.33 | 66.33 | 64.28 | 62.83 | 62.27 | 61.19 | 59.99 | 62.60 | 67.31 |
| 10 | **PIP** | **92.70** | **87.90** | **86.83** | **88.23** | **87.04** | **87.18** | **85.96** | **85.13** | **84.62** | **82.35** | **86.79** |
| 8 | Fed-DualPrompt | 74.80 | 72.05 | 71.60 | 70.08 | 68.14 | 66.72 | 66.63 | 65.80 | 65.57 | 71.81 | 69.32 |
| 8 | **PIP** | **90.10** | **87.10** | **87.17** | **87.53** | **86.16** | **86.33** | **85.23** | **83.98** | **83.06** | **81.64** | **85.83** |
| 6 | Fed-DualPrompt | 73.40 | 73.10 | 72.07 | 70.60 | 69.30 | 67.68 | 66.69 | 67.60 | 66.43 | 71.02 | 69.79 |
| 6 | **PIP** | **91.50** | **87.20** | **87.97** | **89.30** | **87.76** | **87.78** | **86.20** | **85.88** | **84.78** | **82.94** | **87.13** |
| 4 | Fed-DualPrompt | 72.30 | 72.05 | 73.27 | 71.60 | 71.48 | 69.70 | 68.74 | 67.20 | 67.26 | 69.40 | 70.30 |
| 4 | **PIP (Our)** | **90.70** | **87.15** | **87.17** | **87.43** | **86.98** | **86.75** | **85.17** | **84.05** | **83.26** | **82.11** | **86.08** |
| 2 | Fed-DualPrompt | 73.90 | 75.55 | 75.10 | 74.28 | 72.44 | 72.75 | 71.51 | 69.45 | 68.78 | 70.71 | 72.45 |
| 2 | **PIP (Our)** | **89.20** | **87.90** | **87.63** | **88.48** | **87.52** | **86.68** | **84.86** | **83.00** | **83.31** | **82.14** | **86.07** |

Table 16: Complete numerical results on TinyImageNet (T=10, local clients = 3) with smaller rounds in one-seeded run i.e. 2021

