# OpenReview forum: "Prototypes-Injected Prompt for Federated Class Incremental Learning"
_ICLR.cc/2024/Conference — ICLR 2024 Conference Withdrawn Submission_

### Official Review · Reviewer_P3dj · 2023-10-30

**Soundness:** 3 good
**Presentation:** 3 good
**Contribution:** 3 good
**Rating:** 6
**Confidence:** 1

**Summary:**

In this work, the authors focus on the  Federated Class Incremental Learning (FCIL) to address the catastrophic forgetting and non-IID data distribution in continual learning. For this purpose, they proposed a method, Prototypes-Injected Prompt (PIP), which introduces prototype injection, augmentation, and weighted Gaussian aggregation. By comparing this model with the baselines, they showed that the proposed model  outperforms existing methods by 14-33% in CIFAR100, MiniImageNet, and TinyImageNet datasets.

**Strengths:**

1. The authors proposed a prompt-based federated continual learning, in order to solve the FCIL problem.
2. The authors designed a new baseline method for the FCIL problem
3. They conducted extensive experiments on three public datasets and demonstrated the robustness of the proposed method in different task sizes, smaller participating clients, and smaller rounds per task.

**Weaknesses:**

1. I suggest the authors provide some detail information of proposed model, in order to help the reproduction.
2. I suggest the authors give some introduction for the specific task, which may help the readers to understand their work clearly.

**Questions:**

Please refer to weaknesses.

**Details Of Ethics Concerns:**

Nan

---

### Official Review · Reviewer_5UUj · 2023-10-31

**Soundness:** 3 good
**Presentation:** 3 good
**Contribution:** 3 good
**Rating:** 5
**Confidence:** 4

**Summary:**

This article proposes a novel prototypes-injected prompt (PIP) method for the FCIL problem. PIP aggregates only three parameters, prompt, head, and prototypes, which greatly reduces communication overhead while improving model accuracy. Extensive experiments have proven the effectiveness of PIP.

**Strengths:**

The logic of the article is clear, the story line makes sense, and it is also better written.

**Weaknesses:**

This paper lacks model architecture diagrams for specific implementations of PIP and some details need to be refined. In order to make the contributions of the article clearer and easier to understand, the following issues need to be addressed:
1. Based on Fig. 1, a concrete model diagram of the proposed prototypes-injected prompt (PIP) approach should be given in Section IV, incorporating the dual-prompt, prototype augmentation, and server weighted Gaussian aggregation modules.
2. Although the experimental part of the article proves that the performance improvement of PIP is significant, we notice that PIP uses a different backbone than the comparison method, which is very unfair. It is recommended to compare under the same backbone.
3. The readers will be more excited to see more local clients involved than fewer local clients involved. As we konw, in the practical application of FCIL, we always want to solve the problem for more clients. It is recommended to add and analyze the performance of the situation.
4. The model was trained and tested only on smaller datasets. How does it perform on slightly larger datasets, such as ImageNet-R or DomainNet?
5. In real-world FCIL problems, the number of clients and the training ratio of each client is unknown, so whether weighting by ratio is necessary. Also, in the ablation experiments, The weighted aggregation module improves the performance very limited and this module may increase the complexity of the model.
6. Please add the experiments of PIP under three datasets at T=10. Also, the total number of classes for the three datasets in Fig2 is incorrectly labeled, The CIFAR100 and miniImageNet datasets each contain 100 classes.
7. There are a lot of math symbols in the article, so check to make sure that a specific explanation is given for each symbol. For example, w_c2^tfor prototype augmentation in the proposed method, when lowercase w means.
8. In the related work section, GLVC Dong et al. (2022) should be GLFC Dong et al. (2022). Please check carefully in the article for the same error.

**Questions:**

See the above

---

### Official Review · Reviewer_3BAT · 2023-10-31

**Soundness:** 1 poor
**Presentation:** 1 poor
**Contribution:** 1 poor
**Rating:** 3
**Confidence:** 5

**Summary:**

A prototypes-injected prompt method is proposed for rehearsal-free federated class incremental learning. The proposed method involves 3 parts: prototype injection on prompt learning, prototype augmentation, and weighted Gaussian aggregation. The performance is validated on three datasets: CIFAR100, MiniImageNet and TinyImageNet.

**Strengths:**

Prompt learning is introduced into FCIL, which can effectively reduce the training and communication overhead of the large model, so that the large model can be applied to FCIL.

**Weaknesses:**

Idea:

1.The idea of freezing the feature extractor to reduce the communication and training overhead has been proposed in "Federated Reconnaissance: Efficient, Distributed, Class-Incremental Learning", and the method in that article is better than this paper in terms of computational and communication efficiency, so this paper is not innovative enough in terms of saving computational and communication overhead.

Methodology:

1.There is no specific design in the paper to address catastrophic forgetting and non-i.i.d., which are the core challenges of FCIL.

Experiments:

1.Unfair experimental comparison: "PIP and Fed-DualPrompt use pre-trained ViT as the backbone network, while the competitors use LeNet", the difference in the feature extractors may lead to a huge performance gap, and the feature extraction ability of ViT is much better than LeNet, which may be the reason why this paper's method outperforms the other comparative methods.

2.Lack of comparison methods: Most of the "10 state-of-the-art algorithms" mentioned in this paper are comparison methods designed in "Federated Class-Incremental Learning", and this paper does not compare other FCIL methods, such as:

"Federated Continual Learning with Weighted Inter-client Transfer"

"Federated Probability Memory Recall for Federated Continual Learning"

"Continual Federated Learning Based on Knowledge Distillation"

"Better Generative Replay for Continual Federated Learning"

"Federated Continual Learning through Distillation in Pervasive Computing"

"Federated Reconnaissance: Efficient, Distributed, Class-Incremental Learning".

3.Lack of comparison of reasoning efficiency: Client-side reasoning on large models such as "ViT" imposes a large computational overhead, and this paper does not compare the reasoning overheads of different approaches.

Writing:

1.Introduction, paragraph 3, second line: "the performance" should be deleted.

2.Introduction, paragraph 3, line 8: "based on prompt learning inspired by prompt learning" is redundant.

Other:

1.The code in the link in the article contains author information

https://anonymous.4open.science/r/an122pouyyt789/l2plib/continual_datasets/continual_datasets.py

**Questions:**

1.How does this article address catastrophic forgetting as well as non-i.i.d.?

2.How does the inference overhead of this paper's method compare to other comparative methods?

---

### Official Review · Reviewer_Rs3a · 2023-11-01

**Soundness:** 3 good
**Presentation:** 1 poor
**Contribution:** 2 fair
**Rating:** 3
**Confidence:** 4

**Summary:**

This work studies the federated continual learning problem where each client deals with a class incremental learning problem, and the server aggregates all clients via weighted averaging. The authors propose a prompt learning based method that includes prototype injection, prototype augmentation, and weighted Gaussian averaging. Experiments show that their method has significant advantages over existing approaches.

**Strengths:**

* Federated class incremental learning is a challenging but more realistic problem than existing federated learning. Developing exemplar-free methods is an important and interesting direction.

* Experiments show that the proposed approach performs better than other compared baselines.

**Weaknesses:**

* It is acknowledged that prompt learning has been used in class incremental learning, but the reviewer doubts the privacy issue in federated learning setup when using pre-trained ViT as the backbone.
* The ablation study indicates that the prototype strategy impacts significantly the performance. However, the idea of prototype strategy seems not original to me. As far as I know, it has been used in (Dong et al. 2022, 2023) , and prototype augmentation is also known and used in conventional exemplar-free class incremental learning.
* The writing needs to be significantly improved to meet the requirements of a top conference like ICLR. For instance,
>* The cited reference should be given in parentheses in most cases, e.g., using \citep in ICLR latex. The current form of citation makes it hard to read.
>* Every equation (eq. 1-11) in this paper should be followed by punctuation. However, they are missing in this paper.
>* The T^{r,t}_{l}consists of samples in the input space, while prototype sets are in the deep feature space. Therefore, the formulation of the fourth line on page 5 is problematic.
>* In the part of Benchmark Algorithms, the name of each compared method is mixed with the name of the authors of each citation.
>* In the third paragraph of the Introduction section: ..the performance Besides the performance issue,...
>* In eq.6 and eq.7, using the indicative function is better than using “if”, particularly when it is wrongly written in variable form rather than text form.

**Questions:**

* What is the L_{match}?